# Long-term stability of cortical ensembles

**Jesús Pérez-Ortega\*, Tzitzitlini Alejandre-García, Rafael Yuste**

Department of Biological Sciences, Columbia University, New York, United States

**Abstract** Neuronal ensembles, coactive groups of neurons found in spontaneous and evoked cortical activity, are causally related to memories and perception, but it is still unknown how stable or flexible they are over time. We used two-photon multiplane calcium imaging to track over weeks the activity of the same pyramidal neurons in layer 2/3 of the visual cortex from awake mice and recorded their spontaneous and visually evoked responses. Less than half of the neurons remained active across any two imaging sessions. These stable neurons formed ensembles that lasted weeks, but some ensembles were also transient and appeared only in one single session. Stable ensembles preserved most of their neurons for up to 46 days, our longest imaged period, and these 'core' cells had stronger functional connectivity. Our results demonstrate that neuronal ensembles can last for weeks and could, in principle, serve as a substrate for long-lasting representation of perceptual states or memories.

## Introduction

Neuronal ensembles, defined as a group of neurons that fire together, are thought to underlie the neural representations of memories, perceptions, thoughts, motor programs, computations, or mental states (*Lorente de No, 1938*; *Hebb, 1949*; *Cossart et al., 2003*; *Ikegaya et al., 2004*; *Sasaki et al., 2007*; *Buzsáki, 2010*; *Shepherd and Grillner, 2010*; *Yuste, 2015*; *Stringer et al., 2019b*; *Carillo-Reid and Yuste, 2020*). Using two-photon calcium imaging, ensembles have been found in mouse visual cortex during spontaneous activity and after visual stimulation (*Cossart et al., 2003*; *Miller et al., 2014*; *Carrillo-Reid et al., 2015b*; *Stringer et al., 2019c*). The optogenetic activation of the ensembles can lead to behavioral effects consistent with the hypothesis that they represent perceptual or memory states (*Carrillo-Reid et al., 2019*; *Marshel et al., 2019*). Interestingly, while single-cell tuning remains stable in visual cortex (*Ranson, 2017*; *Jeon et al., 2018*), a representational drift occurs across days (*Deitch et al., 2020*). Both stability and flexibility in different brain areas have been reported using single-cell recordings (*Lütcke et al., 2013*; *Ziv et al., 2013*; *Driscoll et al., 2017*; *Gonzalez et al., 2019*; *Rule and Harvey, 2019*). However, there is a lack of multineuronal studies on cortical activity across days. Thus, we asked whether ensembles are preserved across days and how flexible they are, that is, how many neurons firing together on one day continue to do so in the following days and how many of them stop firing together. We also explored whether the stability or flexibility of ensembles are different between spontaneous and visually evoked activity. To study this, we performed longitudinal calcium imaging experiments using two-photon multiplane microscopy in visual cortex of awake mice and measured the responses of the same neurons for up to 46 days. Functional connectivity based on neuronal coactivity was used to detect neuronal ensembles. We found that more than 50 % of ensembles during spontaneous and visually evoked activity are stable. The rest of the ensembles (transient ensembles) appeared in only one session with no difference in the number of neurons or functional structure compared to stable ensembles. Analyzing stable ensembles, we found that ~68 % of their neurons were preserved over weeks (stable neurons), whereas the rest were not (flexible neurons). Functional connectivity analysis revealed that stable neurons were more connected than neurons which were eventually lost. Our results reveal long-term stability, over several weeks, of ensembles built, mostly, by neurons that are more functionally connected.

**\*For correspondence:**
jesus.perez@columbia.edu

**Competing interest:** The authors declare that no competing interests exist.

## Results

### Experimental and analysis rationale

We performed two-photon calcium imaging of pyramidal cells in layer 2/3 of the visual cortex from six transgenic mice (GCaMP6s, n = 4 animals; GCaMP6f, n = 2) through a cranial window to examine the stability of ensembles under visually evoked and spontaneous activity. We head-fixed mice in front of a blue screen monitor, and they were free to run on a wheel (*Figure 1A*). We first measured the spontaneous neuronal activity in response to a static blue screen (*Figure 1B*). Then, we recorded visually evoked activity by displaying 50 times a single-orientation blue drifting gratings stimulus (2 s each) with a static blue screen between presentations (at 1–5 s random intervals; *Figure 1C*). For either spontaneous or evoked activity, we recorded three sessions each day. To track the same neurons across days, multiplane calcium imaging was performed. A reference plane (0 μm) from day 1 was first located, then two extra planes 5 μm apart were recorded, above (–5 μm) and below (+ 5 μm) the reference plane (*Figure 1D*, left). After imaging, maximum intensity projection frames were created from the three planes to assemble a single video per session (*Figure 1D*, right). We then identified regions of interest (ROIs) of neuronal activity and kept neurons with peak signal-to-noise ratio (PSNR) >18 dB (*Figure 1E*, left). Calcium signals from each ROI were then deconvolved for spike inference and thresholded to generate a binarized signal, which we used to build spike raster plots (*Figure 1E*, right). We then analyzed all ROIs to build a binary raster plot (*Figure 1F*) and recorded the activity of the same neurons on days 2, 10, and 43 or 46 (*Figure 1G*). Some animals were imaged on day 43 and others on day 46, but we combined the data from those days (days 43–46; *Figure 1—source data 1*).

We first examined if recording duration influenced the number of active cells found. As neurons can become active at different times, one would expect to capture more active neurons the longer the recording session. At the same time, a very long imaging session would not be practical. To define an imaging duration that significantly captured the neuronal activity present in the imaged field, pilot experiments were carried out and data were tabulated across imaging sessions of increasing duration (*Figure 1—figure supplement 1A*). We found that the accumulated number of active neurons reached a plateau after a few minutes. Based on this curve, we reasoned that, with our imaging and analysis pipeline, intervals of 5 min would capture the majority of active neurons in the imaged territories and carried out the rest of the study by imaging spontaneous and evoked activity during 5 min intervals.

### Less than half of neurons remained active between sessions

We then inquired if the number of active neurons was constant across time and counted active neurons in the field of view on different days. On day 1, we found an average of 83 ± 6 and 85 ± 5 (mean ± SEM) active neurons during spontaneous and evoked activity imaging periods, respectively. While the number of active neurons was similar for days 1–10, a significant decrease in the number of active neurons occurred in days 43–46, in both spontaneous and evoked activity (*Figure 1H*; 51 ± 10 neurons; p=0.023 and p=0.013, respectively). Differences in *z*-displacement were similar across days (<4 μm, see Materials and methods, *Figure 1—figure supplement 2A*), but mice running speed tended to increase across days (*Figure 1—figure supplement 2B*). These factors could not explain the decrease in the number of active neurons. However, the percentage of discarded neurons, with poor signal to noise (PSNR <18 dB), significantly increased from day 1 to days 43–46 (from 24 ± 1 % to 37 ± 4%, mean ± SEM; p=0.034; *Figure 1—figure supplement 2C*). Therefore, the loss of active cells on days 43–46 was likely due to a decrease in imaging quality, which could be partially explained by surgical attachment-related traumas (*Figure 1—figure supplement 3*). At the same time, the percentage of time that a neuron was active (the fraction of frames with activity) did not change over time and was ~15.5 % either for spontaneous or evoked activity (*Figure 1I*). On average, the level of neuronal activity remained similar across days.

We then explored if active neurons from the first session remained active in following sessions (*Figure 1J*). We named 'common' the neurons that were repeatedly active in two different sessions. Surprisingly, less than 50 % of the neurons were repeatedly active across days, and this proportion became reduced over time (*Figure 1K*). Even 5 min later, within day 1, only 42 ± 2 % and 37 ± 4 % (mean ± SEM) of neurons remained active in the subsequent imaging session during spontaneous and evoked activity (gray in *Figure 1K*). This low number of common neurons was not explained by the possibility that not all active neurons are completely captured in a 5 min interval as this number

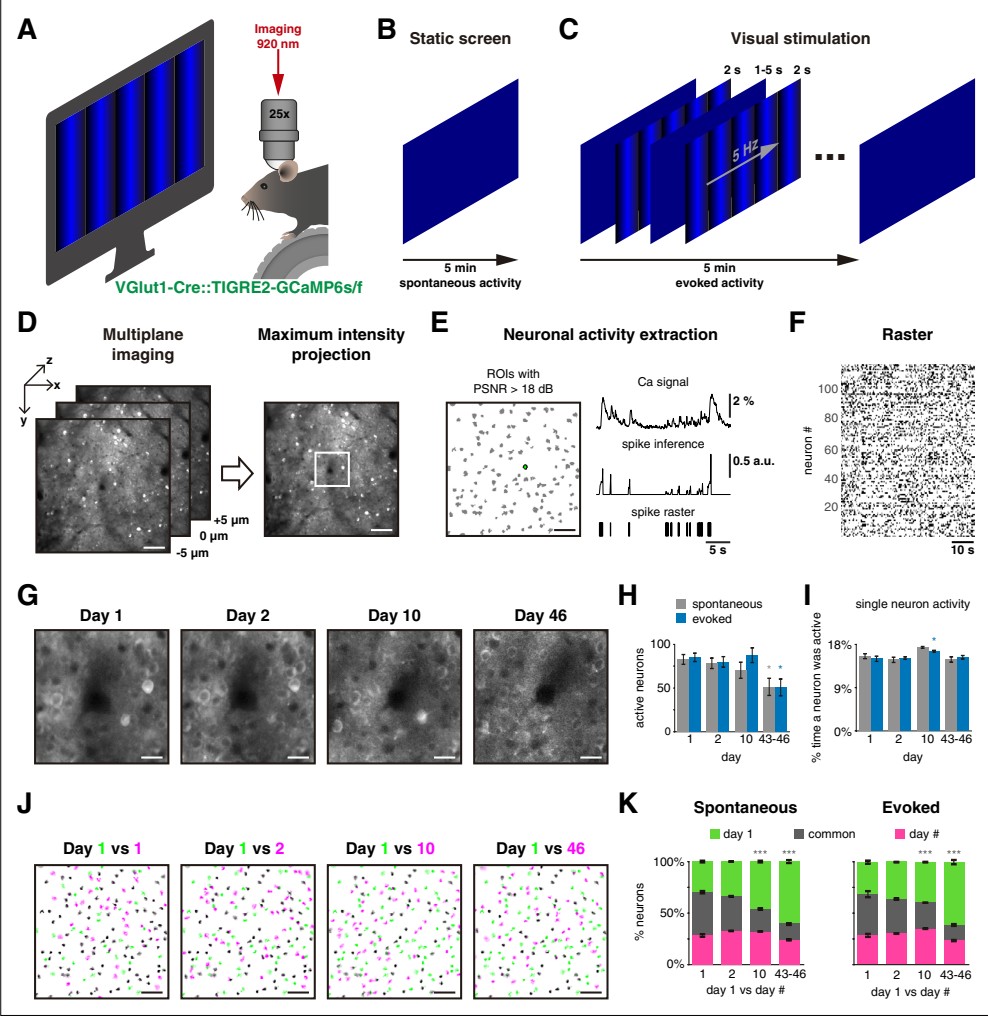

**Figure 1.** Experimental protocol. (**A**) Experimental setup: a mouse with a cranial window is placed on a treadmill in front of a blue screen monitor, and it is head-fixed under the two-photon microscope for calcium imagin. (**B**) Static blue screen was used to record spontaneous activity during 5 min, three sessions per day, 5 min apart between them. (**C**) Visual stimulation protocol constituted of 50 repetitions of a 2 s single-orientation drifting gratings with a mean static screen between each of them during 1–5 s randomly to record evoked activity for 5 min, three sessions per day, 5 min apart between them. (**D**) Strategy to image the same neurons in the field of view on different days: (left) a single plane was carefully located in a reference recorded position (day 1), then two extra planes also were imaged separated 5 μm up and down. Three planes were imaged in a period of 81 ms. (Right) Maximum intensity projection was obtained from the three planes generating a single frame. Scale bar: 50 μm. (**E** left) Detection of regions of interest (ROIs; gray shapes) based on Suite2P algorithm, green ROI is used as a representative example, scale bar: 50 μm; (up right) extraction of calcium signal (ΔF/F$_0$) with peak signal-to-noise ratio (PSNR) >18 dB; (middle right) spike inference using foopsi algorithm; (bottom right) a binary signal obtained by thresholding spike inference, which is used to represent the active frames of the neuron. (**F**) Raster plot built with binary signals from active neurons recorded simultaneously. Each row represents the activity of a single neuron, black dots represent the activity of the neuron. (**G**) Center of an example image at 4× zoom (white square on **D** right) recorded in the same location up to 46 days after the first day of recording. Note that image at day 46 is noisier than on the first days. Scale bar: 12.5 μm. (**H**) Count of active neurons in different days. The number of active neurons identified on day 1 decreased significantly on days 43–46 during spontaneous and evoked activity (p=0.023 and p=0.013, respectively). (**I**) Percentage of single-neuron activity, that is, percentage of frames a neuron was active. The average activity of all neurons was ~15 % on all days during spontaneous and evoked activity. (**J**) Merge of active neuron ROIs from two sessions: first session from day 1 (green in four panels) versus a second single session from same day 1 (5 min later), one session from days 2, 10, and 46 (from left to right, respectively, magenta). The intersection of active neurons in both sessions is in gray color. Scale bar: 50 μm. (**K**) Percentage of active neurons between two sessions: first session from day 1 (green) versus second sessions from days 1, 2, 10, and 46 (magenta).

*Figure 1 continued on next page*

*Figure 1 continued*

Common active neurons (gray) in both sessions were 42% ± 2 % during spontaneous and 37% ± 4 % during evoked activity. There were no significant differences on common active neurons between days 1 and 2, but a significant decrease on days 10 and 43–46 during spontaneous (p=2 × 10$^{-6}$ and p=4 × 10$^{-9}$, respectively) and evoked activity (p=2 × 10$^{-4}$ and p=4 × 10$^{-9}$, respectively). Data are presented as mean ± SEM. Kruskal–Wallis test with post hoc Tukey–Kramer: *p<0.05, **p<0.01 , and ***p<0.001. See *Figure 1—source data 2*.

The online version of this article includes the following figure supplement(s) for figure 1:

**Source data 1.** Mice and recording days.

**Source data 2.** Neuronal activity across days.

**Figure supplement 1.** Active neurons in different session durations.

**Figure supplement 2.** Possible causes of the loss of neurons recorded across days.

**Figure supplement 3.** Chronic cranial window across days.

**Figure supplement 4.** Single-neuron activity and its locomotion correlation across days during spontaneous activity.

**Figure supplement 5.** Single-neuron tuning across days during visually evoked activity.

plateaus (*Figure 1—figure supplement 1B*). However, the percentage of common neurons continued to decrease monotonically across days with significant decrements on days 10 and 43–46 in both spontaneous and evoked activity (gray in *Figure 1K*). Despite the generalized decrease in the number of common neurons, we found a significant number of neuron which were still active 43–46 days later during spontaneous and evoked activity (17 ± 7 and 22 ± 8; mean ± SEM). Moreover, these common neurons had stable responses to locomotion correlation and tuning across days (*Figure 1—figure supplements 4 and 5*) consistent with previous studies (*Ranson, 2017*; *Jeon et al., 2018*). We concluded that the neurons activated spontaneously or by visual stimulation are dynamically changing, and only a small proportion of neurons were repeatedly active across sessions from minutes to weeks.

## Neuronal ensembles identification based on functional connectivity

To further test whether a group of neurons remains firing together in following imaging sessions, we evaluated the common neurons between sessions. If we consider non-common neurons across days, we could conclude that neurons are no longer in the ensembles when, possibly, the neurons were instead silent or out of the field of view due to a displacement in the z-plane. Thus, we focused our analysis on the correlational properties of common neurons by identifying the neuronal ensembles they formed and then evaluating if these ensembles remained across days. To do so, we built binary raster plots of the common neurons (*Figure 2A*) and detected ensembles using their functional connectivity (*Pérez-Ortega et al., 2016*). We identify whether there was a significant functional connection between each pair of neurons to build a functional network graph (*Figure 2B*). Specifically, to identify a significant coactivation between every pair of neurons, we first generated 1000 spike raster surrogates by a random circular shift in time of the active frames (*Figure 2C*, left). Then we tabulated how many times a given pair of neurons were coactive by chance and used a 95 % threshold on the cumulative probability from surrogate coactivations to define a functional connection (*Figure 2C*, right). The functional connections of each neuron were independent of its level of activity ($R$ = 0.001, *Figure 2—figure supplement 1*). We then filtered the raster plot removing the activity without significant coactivations (*Figure 2D*). To explore similarities between coactivations, treating each frame as a vector (one frame bin = 81 ms), we computed the Jaccard similarity between every pair of vectors (*Figure 2D*, bottom). Jaccard similarity indicates the fraction of the same active neurons between two vectors, that is, while a value of 0 means that neurons from one vector are all different from those of the other vector, a value of 1 means that all neurons between both vectors are identical. To detect ensembles, we identified patterns of coactivation (i.e., clusters of vectors) by performing hierarchical clustering of all vectors by single linkage, keeping only the most similar vectors (>2/3 Jaccard similarity, red dotted line in *Figure 2E*). Similar vectors were clustered by Ward linkage using contrast index to determine the number of groups (i.e., neuronal ensembles; *Figure 2F*). Finally, we extracted the neuron identity from the ensembles and built their raster plots and spatial maps (*Figure 2G*; see Materials and methods for details). We identified the ensembles for every single 5 min session and evaluated if they were preserved over time.

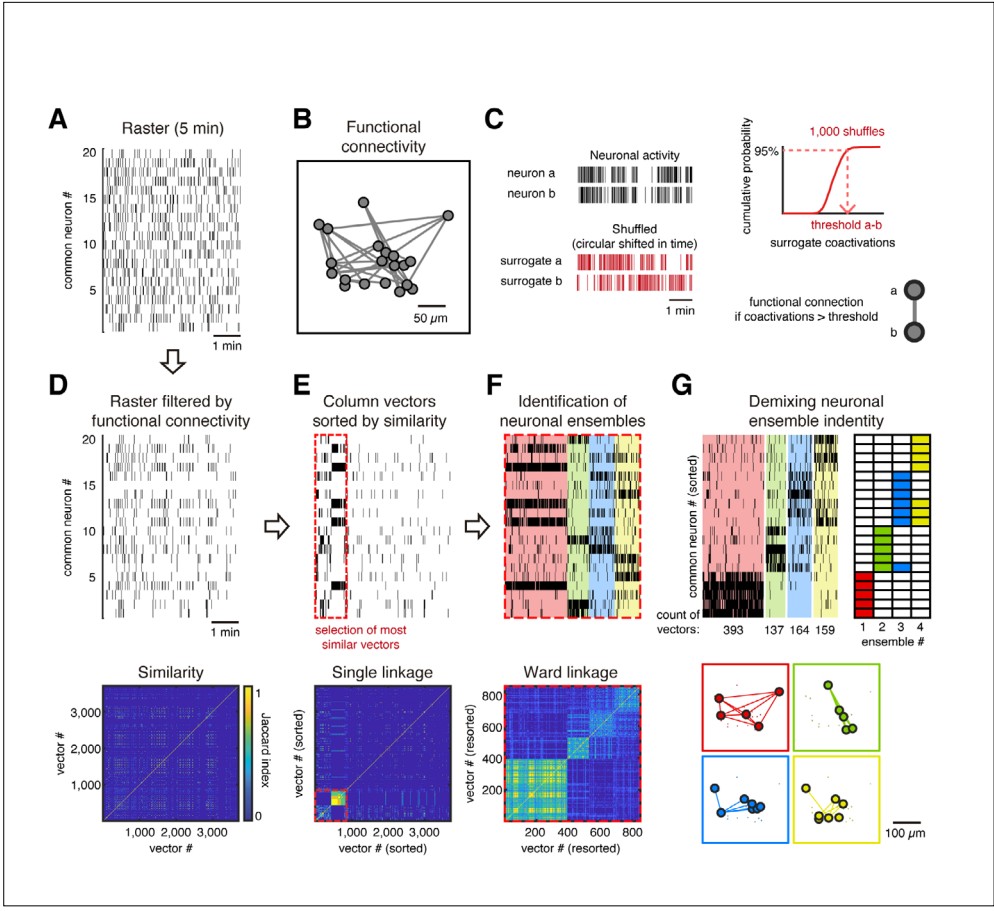

**Figure 2.** Ensemble identification. (**A**) Example of a raster plot from a single session from day 1 of a GCaMP6s mouse during evoked activity showing the common neurons between days 1 and 43. (**B**) Functional neuronal network obtained from raster in (**A**), where every node represents a neuron and every link represents a functional connection between neurons. The network is plotted preserving the spatial location of the neurons. (**C**) Method to identify a significant functional connection between neurons. Taking the activity of each pair of neurons *a* and *b* (top left), we generated 1000 surrogates by a circular shift of their activity (bottom left), in a random amount of time, to disrupt the temporal dependency. Then, a cumulative distribution probability of the surrogate coactivations for each pair of neurons is built (top right), which is used to define a threshold of the number of coactivations at 95 % of chance. Then we put a functional connection between those neurons if they reach a significant number of coactivations (p<0.05), that is, a number of coactivations bigger than the threshold. (**D**, top) Raster (**A**) filtered based on functional connectivity (**B**). We removed activity from neurons with no significant coactivity. (Bottom) Jaccard similarity between every column vector (single frame). (**E**, top) Column vectors from raster in (**D**) sorted by hierarchical clustering using single linkage based on its Jaccard similarity (bottom). Red dotted line indicates the most similar vectors depicted by thresholding the hierarchical clustering with Jaccard similarity >2/3. (**F**, top) Most similar vectors in (**E**) sorted by hierarchical clustering using Ward linkage based on Jaccard similarity (bottom) and grouped in different ensembles (different color each) based on the contrast index. (**G**, top left) Same sorted vectors in (**F**) but here the neurons were sorted depending on their belonging ensemble (top right). (Bottom) Functional neuronal networks representing the ensembles plotted to preserve the spatial location of the neurons.

The online version of this article includes the following figure supplement(s) for figure 2:

**Figure supplement 1.** The number of functional connections of a neuron is independent of its level of activity.

**Figure supplement 2.** Number of ensembles in spontaneous and evoked activity.

## Stable neuronal ensembles can last 46 days

Using this approach, we identified an average of 4.57 ± 0.14 and 4.63 ± 0.14 (mean ± SEM) ensembles on spontaneous and evoked activity sessions, with no significant differences between them (*Figure 2—figure supplement 2*). We then inquired whether ensembles were preserved across days

(stable) or not (transient). To do so, we compared how many neurons of an ensemble captured during the first session were present in an ensemble during a future session. We define a stable ensemble as the ensemble that maintains at least 50 % of its neurons across days (Jaccard similarity ≥1/3; see Materials and methods). This criterion was chosen to set a minimum threshold of similarity to identify an ensemble in a future session, considering the possibility of capturing a rotation in its elements. Comparing two imaging sessions from either spontaneous or evoked activity, we found that some ensembles were preserved but others were not (*Figure 3A,B*). We termed 'stable' the ensembles found on day 1, which were preserved in subsequent days, and 'transient' all other ensembles. On day 1, stable ensembles constituted 54 ± 3 % (mean ± SEM) of all ensembles during spontaneous activity, and 72 ± 4 % (mean ± SEM) of ensembles during evoked activity (*Figure 3C*). Similar trends were observed on days 2 and 10, and stable ensembles were ~50 % of all ensembles in both spontaneous or evoked activity on days 43–46 (*Figure 3C*). Visually evoked ensembles were more stable than spontaneous ones during the first days, but only some of them were similarly stable as spontaneous ones up to 43–46 days. This result suggests that some ensembles during evoked activity could adapt to the stimulus.

To test if stable ensembles were merely an artifact of the analysis, we shuffled the neuronal activity of the session on days 1, 2, 10, and 43 or 46. We found 0.8 ± 0.1 (mean ± SEM) stable ensembles in the shuffled activity compared with 2.8 ± 0.1 (mean ± SEM) stable ensembles from original data ($p=1 \times 10^{-66}$; *Figure 3—figure supplement 1A*). The total time of stable ensemble activation during a single session of spontaneous or evoked activity was significantly higher than in shuffled data (75.1 ± 2.7 s and 14.4 ± 1.3 s, respectively, mean ± SEM, $p=2 \times 10^{-57}$; *Figure 3—figure supplement 1B*). In conclusion, stable ensembles could not be explained by chance, and they were not only reactivated during the following days but also their correlation with locomotion and tuning remained unchanged across days (*Figure 3—figure supplements 2 and 3*). This speaks to the likelihood that neuronal ensembles are true functional circuit elements and not an epiphenomenon of the population activity or a statistical artifact.

## Stable and transient ensembles have similar functional structure but differ in robustness

We also investigated potential differences, other than stability, between stable and transient ensembles, finding similar number of neurons (*Figure 3D*) and network ensemble density (*Figure 3E*). We asked if varying the threshold to define stability could change this result. When we set the strictest threshold (Jaccard similarity = 1, i.e., all the neurons remain in the same ensemble), we counted significantly less stable ensembles ($p<0.05$; *Figure 3—figure supplement 4A*). At the same time, we did not find any significant addition of stable ensembles when we reduced the threshold, even to zero (*Figure 3—figure supplement 4A*). This indicates that the maximum number of stable ensembles could be defined by the minimum number of ensembles found between two sessions. In all cases, regardless of the Jaccard similarity threshold used, there were no significant differences in the functional structure of the ensembles (*Figure 3—figure supplement 4B,C*). This result suggests that the functional structure of the ensembles is constant and independent of its stability. However, ensemble robustness, defined here as the product of ensemble duration and the similarity of its activity (see Materials and methods), was significantly higher in stable than transient ensembles (*Figure 3F*). This result indicates that stable ensembles are more robust than transient ones during spontaneous or evoked activity (*Figure 3—video 1* and *Figure 3—video 2*).

## Stable ensembles are formed by densely connected neurons

Finally, we examined the neuronal identity and functional connectivity of stable ensembles (*Figure 4A–D*). During spontaneous or evoked activity, approximately 50 % of neurons belonged to only one stable ensemble ('single' neurons), while less than 20 % of the neurons belonged to more than one ensemble ('shared' neurons), and the rest of the neurons were not part of any stable ensemble (*Figure 4E*). We inquired what happened to individual neurons of stable ensembles across days. More than 60 % of neurons of a stable ensemble observed on day 1 remained active on future sessions on days 1, 2, 10, and 43–46 during spontaneous and evoked activity ('stable' neurons, *Figure 4F*). The rest of the neurons (<40%) changed to another ensemble or stopped participating in detectable ensembles ('lost' neurons). Interestingly, in subsequent sessions, we found 'new' neurons

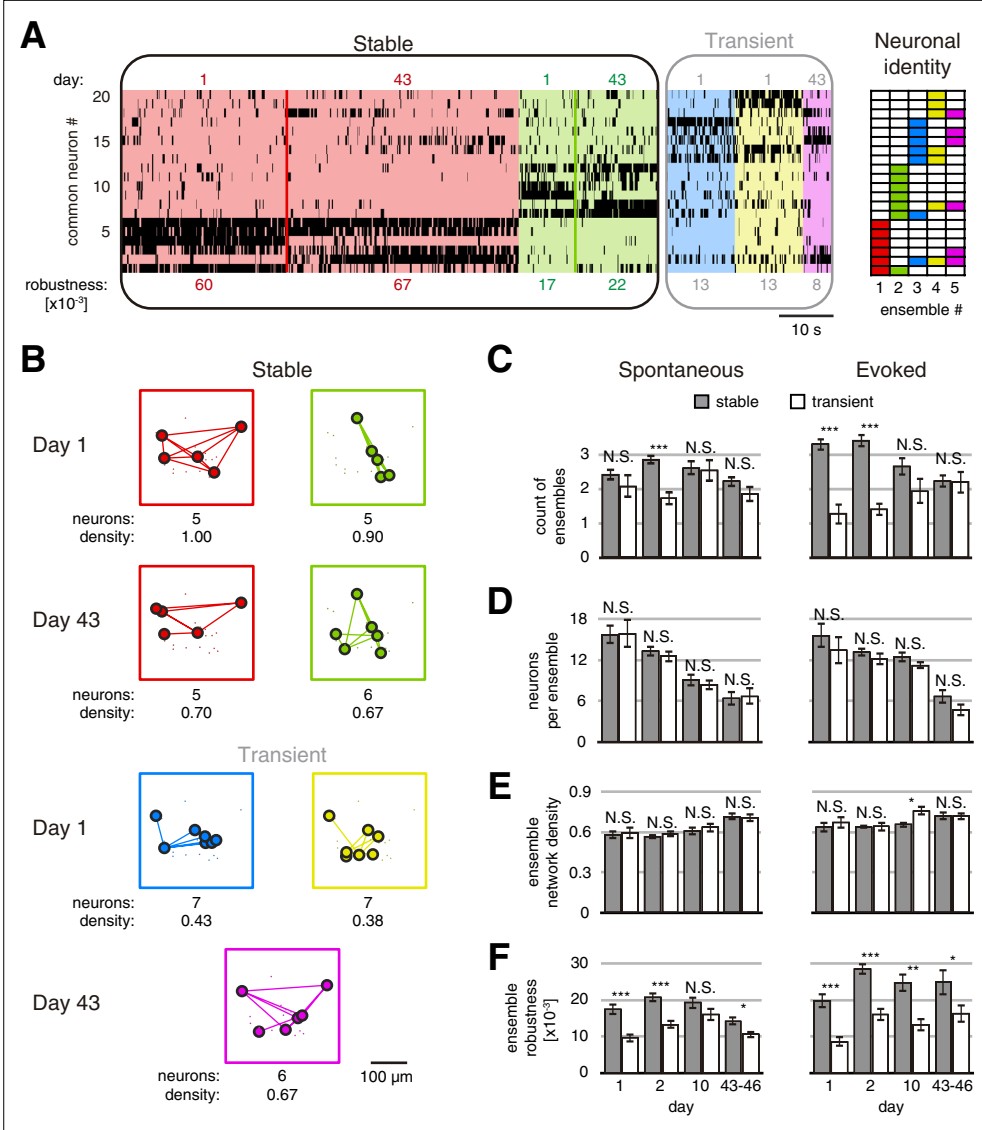

**Figure 3.** Stable and transient ensembles across days. (**A**) Example of raster plot from days 1 and 43 of a GCaMP6s mouse during evoked activity showing the common neurons between days 1 and 43. (Left) Stable ensembles activity sorted by ensemble (background colors show different ensembles) and day (color lines divide the activity from each day in each ensemble). (Middle) Transient ensembles activity sorted by day. (Right) Identity structure of the neuronal ensembles. Ensemble robustness values are at the bottom of each ensemble activity. Ensemble robustness is computed per ensemble per single session. (**B**) Functional networks from ensembles in (**A**) preserving the spatial location of the neurons. Stable (top) and transient (bottom) ensembles separated by day observed. At bottom of each ensemble are values of the number of neurons and the density of the functional connectivity within the ensemble. (**C**) Number of stable and transient ensembles during spontaneous (left) and evoked (right) activity on all days recorded. (**D**) Number of neurons per ensemble with no significant difference between stable and transient ensembles during spontaneous (left) and evoked (right) activity on all days recorded. (**E**) Density of functional ensemble networks had no significant difference between stable and transient ensembles during spontaneous (left) and evoked (right) activity in almost all days, with one exception from evoked activity on day 10. (**F**) Ensemble robustness was significantly higher in stable than transient ensembles during spontaneous activity (left) and evoked (right) activity in all days, with one exception from spontaneous activity on day 10. Mann–Whitney test: *p<0.05, **p<0.01, and ***p<0.001. See *Figure 3—source data 1*.

The online version of this article includes the following video and figure supplement(s) for figure 3:

**Source data 1.** Statistics of stable and transient ensembles.

**Source data 2.** Mice and recording days from Allen Brain Observatory Visual Coding dataset.

*Figure 3 continued on next page*

*Figure 3 continued*

**Source data 3.** Mice from Churchland Lab dataset.

**Figure supplement 1.** Shuffling controls.

**Figure supplement 2.** Correlation of stable ensembles and locomotion across days during spontaneous activity.

**Figure supplement 3.** Stable ensemble tuning across days during visually evoked activity.

**Figure supplement 4.** Ensemble stability analysis varying the threshold to define stable ensembles.

**Figure supplement 5.** Ensemble stability analysis by different methods and datasets.

**Figure 3—video 1.** Stable ensembles in spontaneous activity on days 1 and 46.
https://elifesciences.org/articles/64449/figures#fig3video1

**Figure 3—video 2.** Stable ensembles in evoked activity on days 1 and 46.
https://elifesciences.org/articles/64449/figures#fig3video2

joining stable ensembles in a similar proportion as lost neurons (lost and new neurons, *Figure 4F*). Even when we varied the threshold to define stable ensembles, all of them preserved more than 60 % of their neurons across days (stable neurons, *Figure 4—figure supplement 1B*). Neither stable nor lost neurons specifically belonged to one or more than one ensemble (*Figure 4G*). However, functional connection density between stable neurons from the same ensemble (0.71 ± 0.01, mean ± SEM) was significantly higher than density from lost neurons during spontaneous or evoked activity (0.35 ± 0.02, mean ± SEM, p=7 × 10$^{-55}$, *Figure 4H*). Therefore, weak functional connectivity between lost neurons could explain why they are transient, and high functional connectivity indicates possible lasting stability.

## Ensemble stability is detectable using different methods and datasets

To confirm these results, we performed the same analysis pipeline (*Figure 3—figure supplement 5*; *Figure 4—figure supplement 2*) with our dataset by modifying the way to extract the ensembles and also with two publicly available datasets from the Allen Brain Observatory Visual Coding (*de Vries et al., 2020*) and the Churchland Lab (*Musall et al., 2019*). In our study, we used all single sessions ('main,' three sessions/day/condition with a selection of vectors), and the results were similar when we used only one session per day ('single,' one session/day/condition), all the raster without vector selection ('all vectors'), and the significant population coactivations ('coactivity peaks'; see Materials and methods). The properties of stable and transient ensembles as the neurons/ensemble and connection density remained similar (*Figure 3—figure supplement 5B*,C), but ensemble robustness values showed variability between methods (*Figure 3—figure supplement 5D*). However, the vector selection method used differentiated significantly between stable and transient ensembles, especially during spontaneous activity. In agreement with these conclusions, we found similar results in the Allen Brain Institute dataset during visually evoked activity but a lower number of ensembles and network ensemble density during spontaneous activity (*Figure 3—figure supplement 5A*,C). Although the neurons/ensemble increased after 1 week, network ensemble density and robustness values were consistent with our dataset (*Figure 3—figure supplement 5B*,C). Furthermore, there were no significant differences when we analyzed the Churchland Lab dataset, where evoked activity was relevant for performing a task (*Musall et al., 2019*; *Figure 3—figure supplement 5* and *Figure 4—figure supplement 2*). In all methods and datasets, the number of stable neurons within ensembles was above 60 % (*Figure 4—figure supplement 2B*), the connection density of the stable neurons was above 0.5, which was significantly greater than the connection density of lost neurons (*Figure 4—figure supplement 2D*). Finally, we also observed persistence across days of the temporal sequences of neuronal activations using the seqNMF toolbox (*Mackevicius et al., 2019*; *Figure 4—figure supplement 3*), which could be used in future studies to analyze the dynamic of the temporal structure within ensembles. It should be noted that results were similar between GCaMP6s and GCaMP6f mice, consistent with previous studies (*Musall et al., 2019*). We analyzed four GCaMP6s and two GCaMP6f mice in our dataset, seven GCaMP6f mice from the Allen Brain Institute dataset, and four GCaMP6f mice from Churchland Lab dataset. In summary, we detected long-term ensemble stability regardless of the method or dataset used.

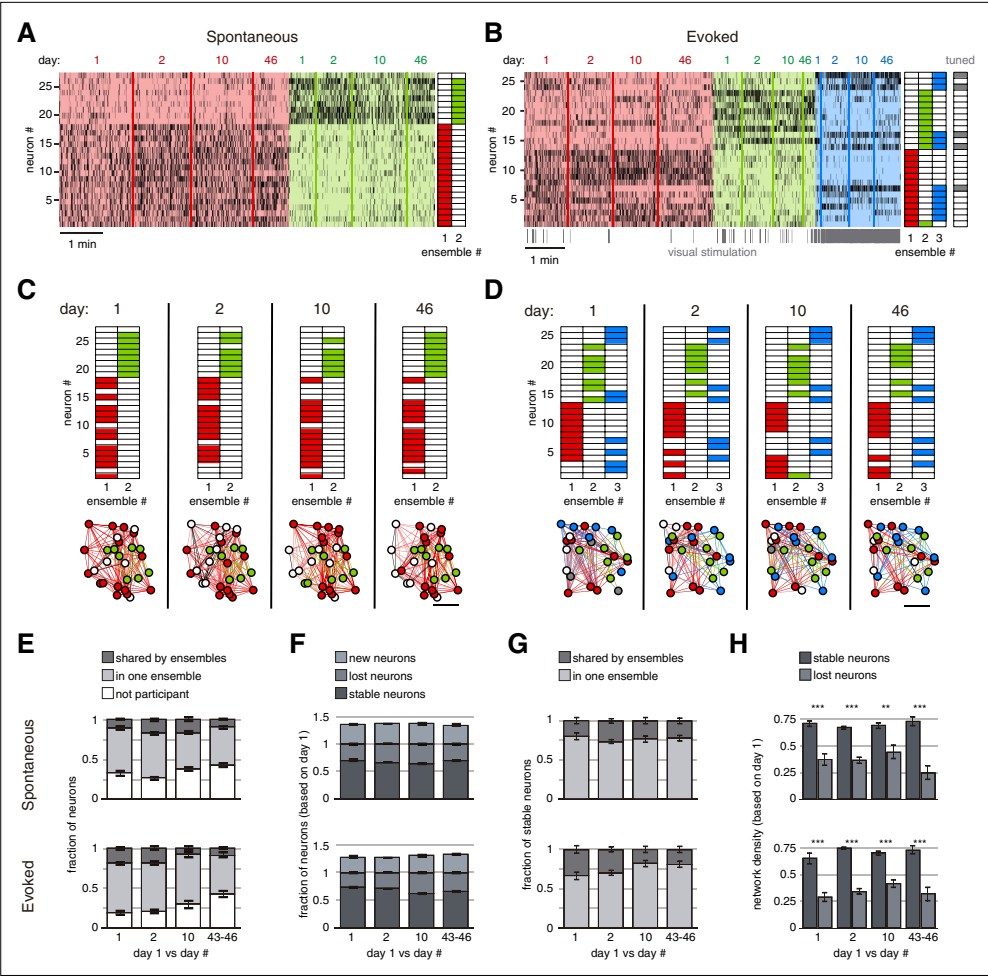

**Figure 4.** Long-term stability of spontaneous and evoked ensembles. (**A**) Example of spontaneous ensemble activity. Neurons are sorted based on their ensemble identity (right). (**B**) Example of evoked ensemble activity. Neurons are sorted based on their ensemble identity, and it is indicated if they were tuned to the visual stimulation (right). The first ~500 ms (of the 2 s) of every visual stimulation (50 per session) are marked at the bottom of the raster activity. Note that there is a particular stable neuronal ensemble (blue ensemble) mainly evoked at the onset of the visual stimulation. (**C**) Neuronal ensembles across days (identified independently). (Top) Neuronal identity sorted as in (**A**). (Bottom) Colors indicate the ensemble to which neurons belonged, white color indicates no participation in any stable ensemble. Scale bar: 50 µm. (**D**) Neuronal ensembles across days (identified independently). (Top) Neuronal identity sorted as in (**B**). (Bottom) Colors indicate the ensemble to which neurons belonged, white color indicates no participation in any stable ensemble, gray color indicates participation in more than one stable ensemble. Scale bar: 50 µm. (**E**) Fraction of neurons during spontaneous (top) or evoked (bottom) activity across days which participated in two or more stable ensembles (shared), only one (single), and without participation in any stable ensemble. (**F**) Fraction of neurons during spontaneous (top) and evoked (bottom) activity across which remained in the same ensemble (stable), changed their ensemble or stopped participating (lost), and new neurons. (**G**) Fraction of stable neurons during spontaneous (top) and evoked (bottom) activity from days 1, 2, 10, and 43–46 which participated in one stable ensemble (single) or more (shared). (**H**) Network density within stable ensembles during spontaneous (top) and evoked (bottom) activity was significantly higher in stable neurons than lost neurons (p<0.01). Density was computed from functional connectivity analyzed on day 1. Data are presented as mean ± SEM. Mann–Whitney test: **p<0.01 and ***p<0.001. See *Figure 4—source data 1*.

The online version of this article includes the following figure supplement(s) for figure 4:

**Source data 1.** Neuronal composition of stable ensembles.

**Figure supplement 1.** Ensemble structure varying the threshold to define stable ensembles.

**Figure supplement 2.** Ensemble structure using different methods and datasets.

**Figure supplement 3.** Persistence of neuronal sequences using seqNMF toolbox.

## Discussion

Starting with Wiesel and Hubel's landmark studies in the 1960s, the plasticity or stability of neuronal activity of the visual cortex has been explored through single-neuron measurements such as receptive field tuning (*Wiesel and Hubel, 1963*; *Wandell and Smirnakis, 2009*; *Clopath et al., 2017*). Recent longitudinal studies have shown that single-neuron tuning is stable for up to 2 weeks (*Ranson, 2017*; *Jeon et al., 2018*). Moreover, single-neuron selectivity is enhanced after learning (*Poort et al., 2015*; *Henschke et al., 2020*). Here, we extend to the microcircuit level these single-neuron longitudinal studies by reporting robust stability in population activity (*Shepherd and Grillner, 2010*; *Yuste, 2015*; *Bargas and Pérez-Ortega, 2017*). We compared, across weeks, microcircuit properties such as functional connectivity, neuronal ensembles, and network topology. In contrast to previous studies, we also measured stability in the absence of visual stimulation. Somewhat surprisingly, we did not find any relevant differences in long-term stability between spontaneous and visually evoked activity. Indeed, we found that ensembles during spontaneous activity were also active during visually evoked activity and vice versa (see 'S day 1 vs. E day n | E day 1 vs. S day n' in *Figure 3—figure supplement 5* and *Figure 4—figure supplement 2*). In fact, more than 60 % of neurons in spontaneous ensembles on day 1 were found to be tuned to a specific stimulus on the following weeks. This result is consistent with the hypothesis that sensory stimuli reactivate existing ensembles, which are already present in the spontaneous activity (*Miller et al., 2014*).

We could not track the activity of every single neuron over weeks with sufficient signal to noise (PSNR >18 dB, *Figure 1H*) since image quality decreased over days (*Figure 1G*). This could be due to the repeated experimental procedures on the same cortical location, decreased transgene expression, laser photobleaching, or surgical-attachment-related microtraumas (*Figure 1—figure supplement 3*). Nevertheless, our method was sufficient to capture similar average activity from the neurons with sufficient levels of signal to noise over weeks (*Figure 1I*), agreeing with the hypothesis that cortical circuits maintain a basic homeostatic activity level, even in spite of perturbations (*Mrsic-Flogel et al., 2007*; *Lütcke et al., 2013*; *Hengen et al., 2013*; *Clopath et al., 2017*). However, we found that single-neuron responses were variable during the same repeated stimulation (*Montijn et al., 2016*; *Stringer et al., 2019a*). Indeed, the chance of finding any given neuron also active in a future session ('common neurons') was less than 60%, even 5 min later (*Figure 1K*; *Tolias et al., 2007*). This result is consistent with transient silencing of neurons (*Prsa et al., 2017*), which could be a neuronal correlate of the learning enhancement in deep neural networks (*Srivastava et al., 2014*; *Rule and Harvey, 2019*). We also found a continuous decrease in 'common' neurons, which could be explained by the loss of neurons (*Figure 1H*) together with small *z*-plane displacement across days (*Figure 1—figure supplement 2A*). If one could maintain imaging quality and focus, we would expect a consistent number of common neurons over weeks, regardless of the time recorded.

Multiplane two-photon calcium imaging allowed tracking the same neurons and identifying neuronal ensembles across days. Stable ensembles were preserved across weeks and 68 % of their neurons continued to be active, while the rest of the neurons were replaced by new ones (*Figure 4F*). The consistent number of neurons preserving their interactions within ensembles across weeks does not support a representational drift in cortical responses at the single neuronal level (*Driscoll et al., 2017*, *Rule and Harvey, 2019*; *Deitch et al., 2020*). In fact, this could be precisely one of the functions of ensembles: to maintain a stable functional state in the midst of an ongoing homeostatic replacement of the activity of individual neuronal elements (*Mrsic-Flogel et al., 2007*; *Lütcke et al., 2013*; *Hengen et al., 2013*; *Clopath et al., 2017*). The stability of ensembles could be based on the stability of dendritic spines (*Yuste and Bonhoeffer, 2001*). The weak connectivity of flexible neurons could be mediated by small spines, which appear and disappear over days (*Holtmaat et al., 2005*), while the high connectivity of stable neurons could be maintained by large spines, which last for months (*Holtmaat et al., 2005*; *Grutzendler et al., 2002*).

One limitation of this study is that we did not image the activity of GABAergic interneurons, which are at least 28 different types based on the morphoelectric and transcriptomic classifications (*Yuste et al., 2020*; *Gouwens et al., 2020*; *Yao et al., 2021*). Parvalbumin (PV) interneurons could stabilize the cortical circuit while somatostatin (SOM) and vasoactive intestinal peptide (VIP) interneurons could modulate the gain of pyramidal neurons (*Bos et al., 2020*; *Millman et al., 2020*). Moreover, single-neuron statistics showed that PVs in the visual cortex undergo faster homeostasis (*Hengen et al., 2013*) and are more stable than pyramidal cells (*Ranson, 2017*). Further studies are needed to

evaluate the stability within interneuron interactions and the interactions between interneurons and pyramidal neurons.

In spite of the incomplete sampling of circuit activity over time and potential alterations on the circuit due to repeated experimental procedures, our results indicate that ensembles can be robust and last several weeks. While there is a significant state of flux in cortical activity at any given moment, there is a subset of neurons that remain active through weeks and form neuronal ensembles. The stability of ensembles that we report could be an underestimate since we only measured snapshots of cortical activity. These stable ensembles, which also have some rotation of their individual neuronal components, appear anchored by core neurons. Specifically, 68 % of neurons remain consistently active within ensembles up to 46 days later (*Figure 4F*) and had stronger functional connectivity (*Figure 4H*). This analysis was robust, even after changing the threshold to define stable ensembles (*Figure 4—figure supplement 1B-D*). These core neurons could be 'anchor' cells, which would maintain stable neural representations and help to maintain them after perturbations (*Rose et al., 2016*; *Clopath et al., 2017*). At the same time, these core neurons could be pattern completion cells, capable of triggering neuronal ensembles (*Carrillo-Reid et al., 2016*; *Carrillo-Reid et al., 2019*). The stronger internal functional connectivity of stable neurons could be mediated by short- or long-term synaptic plasticity (*Carrillo-Reid et al., 2015a*; *Hoshiba et al., 2017*) and may underlie the representation of memories. Since neuronal ensembles in the visual cortex have been associated with perceptual states or memories (*Carrillo-Reid et al., 2019*; *Marshel et al., 2019*), stable neuronal ensembles could represent long-term memories and transient ensembles could illustrate the emergence of new memories or the degradation of existing ones. Future experiments, perhaps using holographic optogenetics (*Yang et al., 2018*) during memory tasks, could test this hypothesis and explore the potential link between the stability of ensembles and the persistence of memories.

# Materials and methods

**Key resources table**

| Reagent type (species) or resource | Designation | Source or reference | Identifiers | Additional information |
|---|---|---|---|---|
| Strain, strain background (mice) | Slc17a7-IRES2-Cre | JAX stock # 023527 | VGlut1 | |
| Strain, strain background (mice) | TIT2L-GC6s-ICL-tTA2 | JAX stock # 031562 | TIGRE2.0 Ai162 | |
| Strain, strain background (mice) | TIT2L-GC6f-ICL-tTA2 | JAX stock # 030328 | TIGRE2.0 Ai148 | |
| Software, algorithm | Drifting gratings generator for visual stimulation | This paper | | https://www.mathworks.com/matlabcentral/fileexchange/78670-drifting-gratings-generator-for-visual-stimulation |
| Software, algorithm | ETL controller for volumetric imaging | This paper | | https://www.mathworks.com/matlabcentral/fileexchange/78245-etl-controller-for-volumetric-imaging |
| Software, algorithm | Catrex GUI | This paper | | https://github.com/PerezOrtegaJ/Catrex_GUI (copy archived at swh:1:rev:2ffc0749535be40ca2331f4c969a82fbfff102d4) |
| Software, algorithm | Neuronal Ensemble Analysis | This paper | | https://github.com/PerezOrtegaJ/Neural_Ensemble_Analysis (copy archived at swh:1:rev:9d37fd031dfbdb4eb69faa449d0a6416267a7d4f) |

## Animals

Experiments were performed on transgenic mice Vglut1 (*Slc17a7*-IRES2-Cre, JAX stock # 023527) crossed with TIGRE2.0 Ai162 (TIT2L-GC6s-ICL-tTA2, JAX stock # 031562) or Ai148 (TIT2L-GC6f-ICL-tTA2, JAX stock # 030328) maintained in C57BL/6 J congenic background. Mice were housed on a 12 hr light-dark cycle with food and water ad libitum. Head-plate procedure and a cranial window were executed after 50 days of age. Mice's health was checked daily. All experimental procedures were carried out in accordance with the US National Institutes of Health and Columbia University Institutional Animal Care and Use Committee.

## Head-plate procedure and cranial window

Adult transgenic mice GCaMP6s (n = 4) and GCaMP6f (n = 2, none had aberrant activity, *Figure 1—figure supplements 2 and 3*; *Daigle et al., 2018*) were anesthetized with isoflurane (1.5–2%). Body temperature was maintained at 37 °C with a heating pad and eyes were moisturized with eye ointment. Dexamethasone sodium phosphate (0.6 mg/kg) and enrofloxacin (5 mg/kg) were administered subcutaneously. Carprofen (5 mg/kg) was administered intraperitoneally. A custom-designed titanium head-plate was attached to the skull using dental cement. Then, a craniotomy was made of 3 mm in diameter with a center at 2.1 mm lateral and 3.4 mm posterior from bregma. A 3 mm circular coverslip was implanted and sealed using cyanoacrylate and cement. After surgery, animals received carprofen injections for 2 days as postoperative pain medication. Mice were allowed to recover for 5 days with food and water ad libitum.

## Visual stimulation

Visual stimuli were generated using a custom-made app on MATLAB (*Pérez-Ortega, 2020a*) displaying on an LCD monitor positioned 15 cm from the right eye at 45° to the long axis of the animal. The red and green channels of the monitor were disabled to avoid light contamination in the imaging photomultiplier (PMT), only the blue channel was enabled (*Kuznetsova et al., 2021*). We used two protocols to display in the monitor. The first was in the absence of visual stimulation, the monitor was displaying a static blue screen, and we used it to record spontaneous activity during 5 min per session. The second protocol was for visual stimulation consisting of full sinusoidal gratings (100 % contrast, 0.13 cycles/deg, 5 cycles/s) drifting in a single direction per mouse (0° or 270°) presented for 2 s, followed by a random amount between 1 and 5 s of mean luminescence. The visual stimulus is presented 50 times during 5 min per session. We performed three consecutive sessions (5 min apart) per protocol per day of the experiment. See *Figure 1—source data 1* for detailed sessions recorded per mouse.

## Multiplane two-photon calcium imaging

Imaging experiments were performed from 20 to 150 days after the head-plate procedure. Each mouse was placed on a treadmill with its head fixed under the two-photon microscope (Ultima IV, Bruker). Animals were acclimated to the head restraint for periods between 5 and 15 min for at least 2 days and exposed to visual stimulation sessions before the recordings presented here. The imaging setup was completely enclosed with blackout fabric to avoid light contamination leaking into the PMT. An imaging laser (Ti:sapphire, $\lambda$ = 920 nm, Chameleon Ultra II, Coherent) was used to excite a genetically encoded calcium indicator (GCaMP6s or GCaMP6f). The laser beam on the sample (30–60 mW) was controlled by a high-speed resonant galvanometer scanning an XY plane (256 × 256 pixels) at 17.7 ms (frame period) covering a field of view of 312 × 312 μm using a 25× objective (NA 1.05, XLPlan N, Olympus). An electrically tunable lens (ETL) was used to change the focus (*z*-axis) during the recording. We recorded consecutively three planes at different depths (–5, 0, and 5 μm from the reference *z*-axis) waiting 9.3 ms between planes for ETL to stabilize the focus. Thus, we collected three frames, one per depth, every 81 ms for 5 min (single session, 3704 frames per plane). Imaging was controlled by Prairie View and ETL was synchronized using a DAQ (USB-6008, NI) controlled by a custom-made app on MATLAB (*Pérez-Ortega, 2020b*).

## Recording same neuronal region through days

On the first day of the experiment, we recorded the vascularization of the pia at 10× and 25× using bright-light microscopy. We fixed the depth to 140 μm from pia to record a reference image (calcium imaging) and a second reference image of the center of the field of view using an extra 4× optical zoom (day 1 in *Figure 1G*). We carefully preserved unchanged the position of the microscope and the base we placed the mice. For the following days of recording, we looked for matching the reference image of vascularization at 10× , then 25× . After that, we looked at 140 μm depth from pia trying to match the reference image on the *x*- and *y*-axis, then we used a 4× extra optical zoom to finely match the second reference image on the *z*-axis (days 2, 10, and 43 or 46 in *Figure 1G*). We performed multiplane imaging to record three planes – one reference plane, one 5 μm above, and one 5 μm below – in order to amend the potential tilt or some *z*-displacement. We evaluated the *z*-plane where neuronal position maximally matched between two sessions and estimated an overall *z*-displacement

of 2.5 ± 0.2, 3.3 ± 0.2, and 3.8 ± 0.1 μm (mean ± SEM on days 2, 10, and 43–46, respectively). The z-displacement difference between days 2, 10, and 43–46 was not significant ($p_{2vs10}$ = 0.83, $p_{2vs43-46}$ = 0.67 and $p_{10vs43-46}$ = 0.98; Kruskal–Wallis test with post hoc Tukey–Kramer). In the end, we extracted the maximum intensity projection from the three planes resulting in a single video for each session (*Figure 1D*).

## Neuronal activity extraction

We used a custom-made graphical user interface (GUI) on MATLAB (*Pérez-Ortega, 2020c*) to extract the binary raster activity from every single session video (5 min, 3704 frames). First, we performed a non-rigid motion correction taking as a reference the mean of the 185 frames (5%) with fewer motion artifacts. Then, we searched the ROIs with a modified version of the Suite2P algorithm (*Pachitariu et al., 2016*, *Figure 1E*). ROIs were preserved if they fulfill the following criteria (fixing radius to 4 μm): $0.5 * \pi * radius^2 < area < 4 * \pi * radius^2$; roundness >0.2; perimeter <$3.5 * \pi * radius$; eccentricity <0.9 and overlapping <60 %. Calcium signal from each ROI was extracted measuring the changes in fluorescence with respect to its local neuropil ($F_{raw}$ — $F_n$)/$F_n$, where $F_{raw}$ is the signal from the ROI and $F_n$ is the signal of its local neuropil 10 times ROI radius. ROI local neuropil is not including the signal from ROIs if presented within the area. Then we computed the PSNR = $20 \cdot log10(max(-F_{raw} — F_n)/\sigma_n)$, where max represents a maximum function and $\sigma_n$ represents the standard deviation of the local neuropil. We evaluated the ROIs again keeping them if PSNR >18 dB. Then we smoothed the calcium signal with a 1 s window average to perform a spike inference using the foopsi algorithm (*Friedrich and Paninski, 2016*). We binarized the spike inference signal, placing 1 if there were spikes inferred and 0 if not. We placed all binarized signals from every ROI in a N × F raster matrix, where N is the number of active neurons and F the number of frames. This matrix is visualized as a raster plot, where the ones in the matrix are the dots representing the active frames of the neurons (*Figure 1F*).

## Tracking neurons across days

We computed a rigid and then a non-rigid motion correction between the binary image of the ROIs shape between a single session of day 1 and a single session from days 1, 2, 10, 43, or 46 (*Figure 1J*). Then, we looked for the intersection (in pixels) between ROIs of the neurons from two sessions (intersection >$0.5 * \pi * radius^2$) and evaluate the Euclidean distance between centroids of the ROIs intersected keeping it if the distance < radius. We used the raster matrix only with the tracked neurons between sessions.

## Identification of neuronal ensembles based on functional connectivity

To analyze neuronal ensembles from raster activity, we used a custom-made GUI on MATLAB (*Pérez-Ortega, 2020d*) Functional connectivity represents the significant coactivity between every pair of neurons from a raster matrix. The number of coactivations $Co_{ab}$ between neuron *a* and *b* was computed counting in how many single frames were both neurons simultaneously active. To identify significance, we generated 1000 surrogates of neurons *a* and *b* by random circular shifting their activity in time to disrupt their temporal dependency. We counted the number of surrogate coactivations $S_{ab,i}$ in each iteration *i*, building a cumulative distribution of $S_{ab}$ selecting a threshold $T_{ab}$ of coactivations at 95 %. If the actual number of coactivations $Co_{ab}$ is above threshold $T_{ab}$, we put a functional connection between neuron *a* and *b* (*Figure 2C*). Doing this with every pair of neurons, we got a functional neuronal network, where every node is a neuron and every link represents a significant coactivity between them (*Figure 2B*). We used the functional connectivity to rebuild the raster matrix to keep the significant coactivity of the neurons. To do so, we identified the active neurons of every single frame and looked at their functional connectivity; if a neuron has no connection, its activity was removed from that frame. At the end, we also removed the frames with less than three coactive neurons (*Figure 2D*). Then we computed the Jaccard similarity between all single frames (column vectors) of the rebuilt raster matrix. A hierarchical clustering tree with a single linkage was obtained to identify the more similar vectors by keeping the branch with more than 2/3 of Jaccard similarity (*Figure 2E*). Using this threshold, we clustered most similar coactivations and filtered non-similar and infrequent coactivations, but similar results were obtained without selecting vectors (all vectors in *Figure 3—figure supplement 5*; *Figure 4—figure supplement 2*). With the more similar vectors, we performed hierarchical clustering with Ward linkage and grouped based on a contrast index (*Beggs*

*and Plenz, 2004*). Each group of column vectors is the activity of the neuronal ensemble (*Figure 2F*), that is, the raster matrix $E_j$ of an ensemble $j$ of size N × $F_j$, where N is the number of neurons and $F_j$ is the number of frames where the ensemble $j$ was active. The time window reported for finding maximum functional coactivity is between 20 and 25 ms (*Buzsáki, 2010*; *Juárez-Vidales et al., 2021*), so one single frame period (81 ms) in our recordings is enough to find ensemble coactivations.

## Detection of coactivity peaks to identify ensembles

Alternatively, we analyzed our dataset using a method to extract neuronal ensembles based on significant population coactivity (*Pérez-Ortega et al., 2016*). Briefly, we obtained a 1 × F vector, where F is the number of frames, by summing the coactive neurons from the raster matrix E. Then, we perform 1000 surrogated raster matrices by randomly circular shifting in time the activity of every single neuron and computing the coactivity given by chance. We determined a significant coactivity threshold (p<0.05) from surrogated coactivity, and the vectors above this threshold were clustered to extract neuronal ensembles.

## Demixing neuronal ensemble identity

The neuronal ensemble activity $E_j$ was used to identify the participation of each neuron in ensemble $j$. We computed the functional connectivity similarly as described above, but incorporating the correlation of the neurons between the times where the ensemble was active. To do so, we got a binary vector $V_j$ representing the times where the ensemble $j$ was active (1) or not (0). Vector $V_j$ was of size 1 × F, where F is the number of frames of the session. A Pearson correlation coefficient $P_{j,a}$ between vector $V_j$ and the activity of neuron $a$ was computed. Then we got an ensemble weight $W_{j,ab}$ between neurons $a$ and $b$ in ensemble $j$, which integrates their correlation with the ensemble $j$ and their number of coactivations as follows: $W_{j,ab} = P_{j,a} \cdot P_{j,b} \cdot Co_{ab}$. To identify significance, we generated 1000 surrogates of neurons $a$ and $b$ shuffling their activity as described before, and assigning randomly a value from the correlation with the ensemble $j$ ($P_j$). Then we compute the surrogate weight $SW_{j,ab,i}$ in each iteration $i$, building a cumulative distribution of $SW_{j,ab}$ selecting a threshold $TW_{j,ab}$ of coactivations at 95 %. If the actual ensemble weight $SW_{j,ab}$ is above threshold $TW_{j,ab}$, we put a functional connection between neurons $a$ and $b$. A neuron is considered to be part of an ensemble if it had at least one single functional connection (*Figure 2G*). A neuron could be part of more than one neuronal ensemble (shared), only one ensemble (single), or in any ensemble (not participant).

## Comparing neuronal ensembles between following sessions

Taking the first session on day 1 and a second session from days 1, 2, 10, 43, or 46, we got the raster matrix from each session with only common neurons (same active neurons in both sessions). Neuronal ensembles were extracted from each raster matrix independently. We define a stable ensemble if 50 % or more neurons matched between an ensemble $j$ found in a first session and a putative same ensemble $j'$ found in a second session. If there is no such match, we called it a transient ensemble. We compute the Jaccard similarity between ensembles $j$ and $j'$, where a value of 0 means that the neurons from $j$ and $j'$ are completely different, and a value of 1 means that neurons from $j$ and $j'$ are exactly the same. We used the value of 1/3 Jaccard similarity as a threshold to keep at least 50 % of the same neurons.

## Ensemble measures

*Ensemble network density*, a fraction of present functional connections to possible connections within an ensemble. *Ensemble robustness*, we introduced here as robustness = *similarity · activity*, where *similarity* is the average of the Jaccard similarity between every pair of column vectors of the ensemble matrix raster $E_j$, and *activity* is the fraction of ensemble active frames to the total frames of the session. The higher the value, the higher the robustness. We computed ensemble robustness for every single session, so it would not be expected beforehand if the ensemble would be stable or not. *Stability of neurons*: given a stable ensemble, a 'stable' neuron participated during the first session on day 1 and a second session on the following days. A 'lost' neuron participated only in the first session but not in a second session, and a 'new' neuron did not participate in the first session but participate in a second session. The fraction is based on total neurons in an ensemble from day 1. *Promiscuity of neurons*: 'shared' neurons is the fraction of neurons participating in more than one

ensemble; 'single' neurons participate in only one ensemble; and 'not participant' neurons do not belong to any ensemble. *Tuned neurons*: we consider a tuned neuron if its number of active frames during visual stimulation was significantly higher than its number of active frames during periods with no visual stimulation ($p < 0.05$, t-test).

### Allen Brain Observatory Visual Coding dataset curation

We found seven mice recorded similarly to our settings: Slc17a7-IRES2-Cre::Camk2a-tTA::Ai93 (Vglut1), VISp structure, and 175 μm depth. The complete methodology can be found in the resource or white paper (*de Vries et al., 2020*; https://observatory.brain-map.org). In brief, two-photon calcium imaging of pyramidal cells in layer 2/3 of the visual cortex from GCaMP6f adult mice was performed in the same region for three different days. Not all mice were recorded in the same sequence of days, so we grouped on days 1, 2–5, and 6–8 (*Figure 3—source data 2*). We downloaded the motion-corrected two-photon calcium fluorescence movies (https://console.aws.amazon.com/s3/buckets/allen-brain-observatory/) and adapted the spatial and temporal resolution to match our movie features. We analyzed the periods of spontaneous activity and visually evoked activity by a natural movie (*Figure 3—figure supplement 5* and *Figure 4—figure supplement 2*). We did not analyze the evoked activity by gratings since they were presented in only one session.

### Churchland Lab dataset curation

The experiments were recording similarly to our dataset setting: Ai93::Emx-Cre::LSL-tTA::CaMK2α-tTA, V1 structure, and 150–450 μm depth. The detailed acquisition of the experiments could be found in *Musall et al., 2019*. In brief, two-photon calcium imaging of pyramidal cells in layer 2/3 of the visual cortex from GCaMP6f adult mice was performed during a visual decision-making task. This dataset had experiments from 10 mice, but only 4 were recorded on V1. Experiments were performed for several days but in different regions, so we analyzed the ensemble stability within 1 day (*Figure 3—source data 3*). We downloaded the 'data.mat' files (http://labshare.cshl.edu/shares/library/repository/38599/) and adapted the temporal resolution to match ours. We created two sessions for each mouse to compare during the same day, and each session was conformed by ~40 continuous trials (~5 min). We analyzed them and compared the ensemble properties during the evoked activity on day 1 (*Figure 3—figure supplement 5* and *Figure 4—figure supplement 2*).

### Finding neuronal sequences using the seqNMF toolbox

We used the seqNMF toolbox (https://github.com/FeeLab/seqNMF; *Mackevicius et al., 2019*) to detect possible neuronal patterns of sequential activation as an alternative to our method, which has a constraint to detect simultaneous neuronal activation (coactivation). We used the common neurons across days during spontaneous and evoked activity to find temporal sequences within windows of ~1 s (L = 12), a regularization parameter lambda = 0.005, and 50 iterations (*Figure 4—figure supplement 3*).

## Acknowledgements

We thank James Holland for his assistance and members of the Yuste Lab for useful comments. This project was supported by R01EY011787 and R01MH115900. RY is an Ikerbasque Research Professor at the Donostia International Physics Center (DIPC). JP has a postdoctoral fellowship from the National Council of Science and Technology from Mexico (CONACYT). The authors have no competing financial interests to declare. JP, TA, and RY conceived the project, planned experiments, and discussed results. TA and JP performed surgeries, JP performed experiments, coded the software, and analyzed the data. JP and RY wrote the paper. RY assembled and directed the team and secured funding and resources. JP dedicates this paper to the memory of Amparo Rodríguez-Cruz.

## Additional information

### Funding

| Funder | Grant reference number | Author |
|---|---|---|
| National Eye Institute | R01EY011787 | Rafael Yuste |
| National Institute of Mental Health | R01MH115900 | Rafael Yuste |
| Consejo Nacional de Ciencia y Tecnología | CVU365863 | Jesús Pérez-Ortega |

The funders had no role in study design, data collection and interpretation, or the decision to submit the work for publication.

### Author contributions

Jesús Pérez-Ortega, Conceptualization, Data curation, Formal analysis, Investigation, Methodology, Software, Writing – original draft, Writing – review and editing; Tzitzitlini Alejandre-García, Conceptualization, Methodology; Rafael Yuste, Conceptualization, Funding acquisition, Project administration, Resources, Writing – review and editing

### Author ORCIDs

Jesús Pérez-Ortega (ID) http://orcid.org/0000-0001-8502-1692
Tzitzitlini Alejandre-García (ID) http://orcid.org/0000-0002-2243-8703
Rafael Yuste (ID) http://orcid.org/0000-0003-4206-497X

### Ethics

All experimental procedures were carried out in accordance with the US National Institutes of Health and Columbia University Institutional Animal Care and Use Committee (protocol AC-AAV3464).

### Decision letter and Author response

Decision letter https://doi.org/10.7554/eLife.64449.sa1
Author response https://doi.org/10.7554/eLife.64449.sa2

## Additional files

### Supplementary files
• Transparent reporting form

### Data availability

Data analyzed during this study are included in the manuscript and supporting files. Links to download the code developed in MATLAB are included in Methods. Data can also be found on Dryad, under the https://doi.org/10.5061/dryad.cfxpnvx5m.

The following dataset was generated:

| Author(s) | Year | Dataset title | Dataset URL | Database and Identifier |
|---|---|---|---|---|
| Pérez-Ortega J, Alejandre-García T, Yuste R, Yuste R | 2021 | Long-term stability of cortical ensembles | https://doi.org/10.5061/dryad.cfxpnvx5m | Dryad Digital Repository, 10.5061/dryad.cfxpnvx5m |

The following previously published datasets were used:

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
