## [Decision Letter]

**Acceptance summary:**

Cortical circuits are subject to ongoing plasticity during an animal's lifetime that has the potential to alter responses to external stimuli over days and weeks. Pérez-Ortega and colleagues examines whether coincident firing of neurons in mouse visual cortex is preserved over a long timescale (one month) in response to repeated stimuli. The authors find that in spite of significant variability in the responsiveness at the population level, subsets of identified neurons maintain coordinated firing. Such cell assemblies may provide a stable substrate for representing visual stimuli.

**Decision letter after peer review:**

Thank you for submitting your article "Long-term stability of cortical ensembles" for consideration by *eLife*. Your article has been reviewed by 3 peer reviewers, and the evaluation has been overseen by a Reviewing Editor and Tirin Moore as the Senior Editor. The following individuals involved in review of your submission have agreed to reveal their identity: Carsen Stringer (Reviewer #1); Laura N Driscoll (Reviewer #2).

The reviewers have discussed the reviews with one another and the Reviewing Editor has drafted this decision to help you prepare a revised submission.

This work examines whether coincident firing of neurons in visual cortex is preserved over a long timescale (one month) which is important because it provides insight into the stability and plasticity of neural circuits and visual representations. The authors find that subsets of identified neurons maintain coordinated firing despite some degree of flux in the firin activity across the population.

All reviewers agreed that the question is important but found the analysis lacked depth and there were some technical issues in the experiments that should be addressed with a fuller discussion and potentially additional analysis to eliminate confounds/artefacts. In general, and in light of earlier work (some of which is not cited) the conclusions need to be more circumspect. Specifically:

– There were concerns about movement/loss of cells/calcium indicator artefacts over this long imaging period that should be accounted for more rigorously.

– The analysis applies a somewhat arbitrary criterion for stability (50% of cells remain in responsive in an assembly). This threshold should be systematically explored and justified more carefully.

– The wider literature on this topic should be more thoroughly cited, limitations of the study should be transparently laid out, claims about the overall stability found in this population response and its relevance to memories and behaviour should be moderated in line with the comments below.

*Reviewer #1 (Recommendations for the authors):*

Perez-Ortega and colleagues performed rigorous experiments to determine if the activity of neurons in visual cortex is similar across days, in particular comparing spontaneous activity in the absence of visual stimuli across days, which was previously not examined to my knowledge. The paper claims that evoked ensembles are more stable than spontaneous ensembles, but more convincing quantitative analyses are required to support these claims.

– There is only one mention of prior work with multi-day imaging in visual cortex (Ranson 2017). Another related study to cite and compare your results to would be Jeon, …, Kuhlman 2018 (and I think a comment about how similar/different your results are from this study + Ranson would be useful for the reader). I would also recommend mentioning that there are studies that have observed differences in evoked activity across learning in V1 (e.g. Poort, Khan et al., 2015; Henschke, Dylda et al., 2020). Do you think there was adaptation across days to the stimulus that you repeated?

– Some GCaMP6f mice have aberrant cortical activity (https://www.ncbi.nlm.nih.gov/pmc/articles/PMC5604087/). In the raw data (Figure 1F) it doesn't look present, but it would be useful to show more time and sort the neurons by their first PC weights perhaps to see the activity structure.

– The approach of 3 plane imaging taking the maximum projection seems useful for tracking cells across days. There is a claim that some cells are no longer found / no longer active. Based on Figure 1G it appears there may have been some Z-movement from day 10 to day 46. This Z movement may explain some of the lost active cells. As a sanity check I would recommend plotting the Z-plane on which the cells were maximally active on day 1 vs the Z-plane on which the cells were maximally active on day n.

– There is an emphasis on analyzing the data as ensembles but I think this may be missing other slow, gradual changes. The definition of stable is at least 50% of neurons were preserved across days. However, the fitting procedure of finding ensembles may produce different ensembles even if those neurons are still correlated to each other. I would recommend two possible additional analyses: (1) compare the correlation matrices for common neurons across days (unless there are too few neurons for this); (2) look at changes in single neuron statistics across days. For (2) this may include reliability of neural responses to the visual stimuli, the weights of the neuron onto the first principal component of spontaneous activity, or the correlation of a neuron with running speed. I think these results may solidify your ensemble result (evoked-related statistics change less across time).*Reviewer #2 (Recommendations for the authors):*

Overall I think the authors collected an interesting dataset. Analyses should be adjusted to include all cells rather than sub-selecting for stability. Additionally, the language needs to be adjusted to better reflect the data. I wish there was any behavioral data included, but if the authors compare their data to publicly available data in V1 for a single recording session during a visually guided task, these concerns could be quelled a bit. Reject, but I would reconsider if the following suggested changes are made.

1. In general the language of this paper and title seem to mismatch the results. The fraction of cells that were 'stable' as the authors say on line 112 was very small, however the authors focus extensively on this small subset for the majority of analyses in the paper. Why ignore the bulk of data (line 119)? What happens if you repeat the same analysis and keep all cells in the dataset? The general language around stability of neural ensembles should be adjusted to better reflect the data (ex: lines 157, 225).

2. There are claims in this paper about how ensembles 'implement long-term memories' in the introduction and conclusion and yet the authors never link the activity of ensembles to any behavioral or stimulus dependent feature. This language reaches far beyond the evidence provided in this paper. The introduction could provide some better framing for expectations of stability vs. drift in neural activity rather than focus on the link between ensembles and memory given that there isn't much focus on the ensembles' contribution to memory throughout. For example, the last sentence of the paper is not supported by data in the paper. Where is the link between ensembles and memory in the data? What is the evidence that transient ensembles are related to new or degraded memories? This reads as though it was the authors' hypothesis before doing the experiments and was not adjusted in light of the results.

3. There is no discussion around the alternative to stability of neuronal ensembles. What are the current theories about representational drift? For example, in Line 34 the authors present an expectation for stability without any reasoning for why there need not be stability. This lack of framing makes their job of explaining results in line 217 more difficult. There is a possibility that the most stable cells aren't more important – what is the evidence that they are? Does an ensemble need a core? Would be interesting to include some discussion on the possibility of a drifting readout (Line 223). [https://doi.org/10.1016/j.conb.2019.08.005]

4. How do activations in V1 in this dataset compare to other data collected from V1 while the animal is performing a task (where for example the angle of the gradings is relevant to how the mouse should respond)? I would be interested to know if the authors compared statistics of their ensembles to publicly available data recorded in V1 during a visually guided behavior. Are the ensembles tuned to anything in particular? Could they be related to movement? [http://repository.cshl.edu/id/eprint/38599/]

5. The authors provide some hypotheses as to why fewer cells are active in the later imaging sessions (dead/dying cells?). This is worrisome in regards to how much it might have affected the imaged area's biology. One alternative hypothesis is that the animal is more familiar with the environment/ not running as much etc. Have the authors collected any behavioral data to compare over time?

6. How much do the results change when you vary the 50% threshold of preserved neurons within an ensemble (Line 146)? Does it make sense to call an ensemble stable when 50% of the cells change? Especially given that the cells analyzed as contributing to an ensemble are already sub-selected to be within the small population of stable cells (Line 119)?

7. Cells are referred to as 'stable' when they're active on 3 different sessions that are separated in time. However, the authors find a smaller number of cells are stable over extended time (43-46 days later). If we extrapolate this over more time, would we expect these cells to continue to be stable? Given these concerns, it might make more sense to qualify the language around stability by the timespan over which these cells were studied.

8. Filtering frames to only coactive neurons for ensemble identification seems strange to me. Authors may be overestimating the extent of coactivation. What happens when you don't do this? How much do the results change when you don't subselect for Jaccard similarity? I would be interested to see how the results vary as you vary this threshold (Line 136).

9. The term 'evoked activity' is misleading because the authors don't link these activations to the visual stimulus. There's no task, so the mice could be paying little attention to the stimulus. Should we really consider this activity to be visually driven? Could the authors provide any evidence of this?

10. A method like seqNMF could reveal ensembles that are offset in time. This looser temporal constraint could potentially reveal more structure. This should be run on the entire dataset (without stability sub-selection). I suggest this as a potential alternative or supplement to the method described by the authors. [https://elifesciences.org/articles/38471]*Reviewer #3 (Recommendations for the authors):*

Neuronal ensembles have been shown by this lab and others to constitute one basic functional unit for the representation of information in cortical circuits. It is therefore important to determine how stable these blocks of representation might be. If these ensembles were preserved across time and sensory stimuli, this would indicate a significant degree of structure underlying cortical representations. In a first attempt to address these important issues, this manuscript analyzes the long-term stability of ensembles of coactive neurons in the layer 2/3 of mouse visual cortex across several days. Ensembles were recorded during periods of spontaneous activity as well as during visual stimulation (evoked). For this, the authors record spontaneous and evoked activity using two-photon calcium imaging one, ten and 40 days after the first recording session. In order to maximize overlap between successive imaging sessions, the authors record three planes separated by 5 microns almost simultaneously (9ms interval) using an electrically-tunable lens. They show that ensembles extracted during visual stimulation periods are more stable on days 2 and 10 than those computed during spontaneous activity. Stable ensembles display a higher "robustness" (a parameter that quantifies how many times a given ensemble is repeated and how similar these repeats are). Neurons displaying stable membership are more functionally connected than unstable ones. It is concluded that such observed stability of spontaneous and evoked ensembles across weeks could provide a mechanism for memories. Long-term calcium imaging within the same population of neurons is a real challenge that the authors seem to overcome in the study. The conclusions are important, my main concern relates to the number of experiments and analyses supporting these findings as detailed below.

Number of experiments and statistics: According to Table 1, two mice with GCamP6f have been through the complete imaging protocol (days 1,2, 10 and 43) but none with the 6s, since 3 missed the intermediate measure (day 10) and one the last point (day 40+). Therefore five mice have been recorded over weeks with two different indicators, but only two were sampled on day 10. One mouse was only recorded until day 10. Altogether, this is quite a low sampling, but the experiments are certainly difficult. However, the total number of experiments analyzed is higher, due to the repeat of 3 sessions on the same mouse per day. This certainly contributes to reaching significance. However, the three samples from the same mouse are not independent points. Are the FOVs different for each session in the same mouse? If they are the same, then the statistics should be repeated but treating all experiments from the same mouse as single experiments. I would suggest repeating the analysis but using only one data point per mouse per day. Also, given that two different indicators were used (6s and 6f), one would need to see whether the statistics are the same in the two conditions.

Robustness: the authors compute this metric, as the product of ensemble duration and average of the Jaccard similarity and find that stable ensembles display higher robustness: isn't it expected that robustness is higher in stable ensembles given that stable ensembles should be observed more often?

Evoked ensembles: It seems to me that evoked ensembles are ensembles extracted during continuous imaging periods that include stimulation. However, one would expect evoked ensembles to be the cells activated time-locked to the visual stimulation. This notion only appears at the end of the paper with "tuned" neurons in Figure 4. In the discussion, authors conclude lines 205-207 that "sensory stimulus reactivate existing ensembles". I do not think this is supported by the analysis performed here. For this, I believe that one would need to compare, within the same mouse the amount of overlap between spontaneous ensembles and "tuned neurons".

How representative are the illustrated examples in Figures 2 and 3? The authors report that about 20 neurons remain active from day 1 to 46 but their main figures display example rasterplots with more than 60 neurons, which is three times more than the average. Is this example representative? Which indicator was used? Is there a difference in stability between 6f and 6s?

Rasterplot filtering: The authors chose to restrict their ensemble analysis to frames with "significant coactivation". Why not use a statistical threshold to determine the number of cells above which a coactivation is significant instead of arbitrarily setting this number to three coactive neurons? In cases of high activity this number may be below significance.

Demixing neuronal identity: The authors assign a neuron to an ensemble if it displays at least a functional connection with another neuron. They use reshuffling to test significance of functional links but still it seems that highly active neurons are more likely to display a high functional connectivity degree and therefore to be stable members of a given ensemble with that definition of ensemble membership. What is the justification to define membership based on pairwise functional connectivity? The finding that core ensemble members display a high functional degree may be just a property reflecting a property of highly active neurons (as previously described by Mizuseki et al., 2013).

Type of neurons imaged: The authors use Vglut1-Cre mice, therefore they are excluding GABAergic cells from their study, this should be clearly mentioned and even discussed.

Volumetric imaging: I am not sure one can say that "volumetric imaging" was performed here, rather this is multi-plane imaging.

Mouse behavior: there is little detail concerning mouse behavior, are mice allowed to run? What is the correlation between ensemble activation and running?

Abstract: the authors should say that 46 days is the longest period they have been recording, otherwise it gives the wrong impression that after 46 days ensembles are no longer stable. Also "most visually evoked ensembles" should be replaced by "ensembles observed during periods of visual stimulation" (see above). "In stable ensembles most neurons still belonged to the same ensemble after weeks": how could ensembles be stable otherwise?

Discussion: I found the discussion quite succinct. It lacks discussing circuit mechanisms for assembly stability and plasticity (role of interneurons for example?), limitations and possible biases in the analysis and placing results in the perspective of other studies analyzing the long-term stability of neuronal dynamics.

---

## [Author Response]

This work examines whether coincident firing of neurons in visual cortex is preserved over a long timescale (one month) which is important because it provides insight into the stability and plasticity of neural circuits and visual representations. The authors find that subsets of identified neurons maintain coordinated firing despite some degree of flux in the firin activity across the population.All reviewers agreed that the question is important but found the analysis lacked depth and there were some technical issues in the experiments that should be addressed with a fuller discussion and potentially additional analysis to eliminate confounds/artefacts. In general, and in light of earlier work (some of which is not cited) the conclusions need to be more circumspect. Specifically:– There were concerns about movement/loss of cells/calcium indicator artefacts over this long imaging period that should be accounted for more rigorously.– The analysis applies a somewhat arbitrary criterion for stability (50% of cells remain in responsive in an assembly). This threshold should be systematically explored and justified more carefully.– The wider literature on this topic should be more thoroughly cited, limitations of the study should be transparently laid out, claims about the overall stability found in this population response and its relevance to memories and behaviour should be moderated in line with the comments below.Reviewer #1 (Recommendations for the authors):Perez-Ortega and colleagues performed rigorous experiments to determine if the activity of neurons in visual cortex is similar across days, in particular comparing spontaneous activity in the absence of visual stimuli across days, which was previously not examined to my knowledge. The paper claims that evoked ensembles are more stable than spontaneous ensembles, but more convincing quantitative analyses are required to support these claims.

We are grateful to this reviewer for the careful and thoughtful constructive comments.

– There is only one mention of prior work with multi-day imaging in visual cortex (Ranson 2017). Another related study to cite and compare your results to would be Jeon, Kuhlman 2018 (and I think a comment about how similar/different your results are from this study + Ranson would be useful for the reader). I would also recommend mentioning that there are studies that have observed differences in evoked activity across learning in V1 (e.g. Poort, Khan et al., 2015; Henschke, Dylda et al., 2020). Do you think there was adaptation across days to the stimulus that you repeated?

We now cite Ranson 2017 and Jeon et al., 2018 as well and we agree with the reviewer’s comment. We highlight that our study differs from previous studies because of its multineuronal level of analysis. We measured the locomotion correlation and the tuning of the common neurons across days and the results were similar to those of previous studies. We have added two supplementary figures with analysis (Figure 1 – Supplement Figure 4-5). Please see lines 119-121 in the Results of the revised manuscript:

“Moreover, these common neurons had stable responses to locomotion correlation and tuning across days (Figure 1 — Figure supplement 4-5) consistent with previous studies (Ranson, 2017; Jeon et al., 2018).”

Also, we now cite Poort et al., (2015) and Henschke et al., (2020) and discuss their contributions. Please see lines 264-270 in the Discussion of the revised manuscript:

“Recent longitudinal studies have shown that single-neuron tuning is stable for up to two weeks (Ranson, 2017; Jeon et al., 2018). Moreover, single-neuron selectivity is enhanced after learning (Poort et al., 2015; Henschke et al., 2020). Here, we extend to the microcircuit level these single-neuron longitudinal studies, by examining the stability of neuronal population (Shepherd and Grillner, 2010; Yuste, 2015; Bargas and Pérez-Ortega, 2017). We compared, across weeks microcircuit properties such as functional connectivity, neuronal ensembles, and network topology.”

We have added as well that there is a possible ensemble adaptation during evoked activity. Please see lines 170-173 in the Results:

“Visually-evoked ensembles were more stable than spontaneous ones during the first days, but only some of them were similarly stable as spontaneous ones up to 43-46 days. This result suggests that some ensembles during evoked activity could be adapted to the stimulus that was repeated.”

– Some GCaMP6f mice have aberrant cortical activity (https://www.ncbi.nlm.nih.gov/pmc/articles/PMC5604087/). In the raw data (Figure 1F) it doesn't look present, but it would be useful to show more time and sort the neurons by their first PC weights perhaps to see the activity structure.

We appreciate this query. Despite the fact that our VGlut1-Cre::Ai148 mice are a different strain from those described by Steinmetz et al., (2017), it still possible that they present aberrant activity due to GCaMP6f expression (40%; Daigle et al., 2018). However, the GCaMP6f mice analyzed in this study did not present aberrant cortical activity. The new two supplement figures (see the previous comment 1) illustrate normal neuronal activity in these GCaMP6f mice (Figure 1 —figure supplement 4-5).

We also have added this clarification in the Methods (lines 355-356 of the revised manuscript):

Adult transgenic mice GCaMP6s (n = 4) and GCaMP6f (n = 2, none had aberrant activity, Figure 1 – Supplement Figure 4-5; Daigle et al., 2018)…”

– The approach of 3 plane imaging taking the maximum projection seems useful for tracking cells across days. There is a claim that some cells are no longer found / no longer active. Based on Figure 1G it appears there may have been some Z-movement from day 10 to day 46. This Z movement may explain some of the lost active cells. As a sanity check I would recommend plotting the Z-plane on which the cells were maximally active on day 1 vs the Z-plane on which the cells were maximally active on day n.

We appreciate this comment. We have performed the suggested analysis by the reviewer and added the z-displacement from day 1 to the following days. As suggested by the reviewer, we estimated the displacement identifying where neurons were maximally matching between two sessions. We observe no significant displacement between days, so this does not explain the monotonic decrease in neuron number across days. We have added the following paragraph in the Methods (lines 409-416 of the revised manuscript):

“We performed multiplane imaging to record 3 planes: one reference plane, one 5 µm above, and one 5 µm below in order to amend the potential tilt or some z-displacement. We evaluated the z-plane where neuronal position maximally matched between two sessions and estimated an overall z-displacement of 2.5 <milestone-start />±<milestone-end /> 0.2, 3.3 <milestone-start />±<milestone-end /> 0.2, and 3.8 <milestone-start />±<milestone-end /> 0.1 µm (mean <milestone-start />±<milestone-end /> SEM on days 2, 10, and 43-46, respectively). The z-displacement difference between days 2, 10, and 43-46 was not significant (p2vs10 = 0.83, p2vs43-46 = 0.67 and p10vs43-46 = 0.98; Kruskal-Wallis test with post hoc Tukey-Kramer). In the end, we extracted the maximum intensity projection from the 3 planes resulting in a single video for each session (Figure 1D).”

However, we counted the number of neurons discarded due to poor signal to noise (< 18dB), and we found a significant increase of discarded neurons only from day 1 to day 43-46 (24 ± 1% and 37 ± 4%, respectively; p = 0.034). This indicated that the loss of active neurons is mainly due to decrease in imaging quality due to lesser SNR, perhaps due to decrease expression of experimentally-related microtraumas.

We added the following paragraph in the Results (lines 93-101 of the revised manuscript):

Differences in z-displacement were similar across days (< 4 µm, see Methods, Figure 1 — Figure Supplement 2A), and mice running speed tended to increase across days (Figure 1 — Figure Supplement 2B). Both factors could not utterly explain the decrease in the number of active neurons. Moreover, the percentage of discarded neurons, with poor signal to noise (PSNR < 18 dB), significantly increased from day 1 to day 43-46 (from 24 ± 1% to 37 ± 4%, mean ± SEM; p = 0.034; Figure 1 — Figure Supplement 2C). Probably, the loss of active cells on day 43-46 was due to a decrease in imaging quality, which could be partially explained by surgical-attachment-related traumas (Figure 1 — Figure supplement 3)”

We added two supplements (Figure 1 — Figure Supplement 2-3).

– There is an emphasis on analyzing the data as ensembles but I think this may be missing other slow, gradual changes. The definition of stable is at least 50% of neurons were preserved across days. However, the fitting procedure of finding ensembles may produce different ensembles even if those neurons are still correlated to each other. I would recommend two possible additional analyses: (1) compare the correlation matrices for common neurons across days (unless there are too few neurons for this); (2) look at changes in single neuron statistics across days. For (2) this may include reliability of neural responses to the visual stimuli, the weights of the neuron onto the first principal component of spontaneous activity, or the correlation of a neuron with running speed. I think these results may solidify your ensemble result (evoked-related statistics change less across time).

We thank the reviewer for this recommendation. We have performed three additional analyses with different criteria as to what represents an ensemble. As suggested by the reviewer, we measured the change in correlation between common neurons across days, which remains around zero across days with no significant difference.

We also performed single-neuron changes in correlation with running speed and in response to the visual stimulation across days (please see Figure 1 —figure supplement 2-3, see previous comment 1).

Additionally, we have compared how stable ensembles change in correlation with running speed and in response to visual stimulation across days. We added two extra supplements (Figure 3 —figure supplement 2-3).

In all cases, the changes were insignificant. Therefore, stability of common neurons is confirmed and extended to longer periods of analysis (46 days) than previous studies (14 days; Ranson, 2017; Jeon et al., 2018). However, we now also show the stability at the multineuronal level, i.e., stable ensembles preserved their correlation with locomotion and tuning with the visual stimulation.

Reviewer #2 (Recommendations for the authors):Overall I think the authors collected an interesting dataset. Analyses should be adjusted to include all cells rather than sub-selecting for stability. Additionally, the language needs to be adjusted to better reflect the data. I wish there was any behavioral data included, but if the authors compare their data to publicly available data in V1 for a single recording session during a visually guided task, these concerns could be quelled a bit.

We appreciate the conscientious review and helpful observations on our work.

1. In general the language of this paper and title seem to mismatch the results. The fraction of cells that were 'stable' as the authors say on line 112 was very small, however the authors focus extensively on this small subset for the majority of analyses in the paper. Why ignore the bulk of data (line 119)? What happens if you repeat the same analysis and keep all cells in the dataset? The general language around stability of neural ensembles should be adjusted to better reflect the data (ex: lines 157, 225).

Thank you for raising an important point that we failed to clarify. We focused the study on the stability of neuronal ensembles, i.e., we tried to test whether a group of neurons that fire together on one day will be firing together in the following days. The null hypothesis is that coactive neurons will not be coactive in followings sessions. To test it, we need to define the “common neurons”, i.e., neurons active in different sessions. If we keep all neurons, we cannot validate or falsify our hypothesis, e.g., in a case without any common neurons between two sessions. Hence, we focus on common neurons. We have clarified in the Results (lines 124-130 of the revised manuscript):

“To further test whether a group of neurons is firing together in the following sessions, we necessarily need to evaluate the common neurons between sessions. If we consider the non-common neurons across days, we could misunderstand that neurons are no longer in the ensembles when, possibly, the neurons are silent or out of the field of view due to a displacement in the z-plane. Thus, we focused our analysis on the correlational properties of the common neurons by identifying the neuronal ensembles they formed and then evaluating if these ensembles remained across days.”

Also, we clarified the text in the Results (lines 117-119 of the revised manuscript) to avoid misunderstandings when referring to common neurons, i.e., active neurons in both sessions. We edited the following paragraph:

“We concluded that the neurons activated spontaneously or by visual stimulation are dynamically changing, and only a small proportion of neurons were repeatedly active across sessions from minutes to weeks.”

However, we agree to adjust the language describing stability and also mention flexibility. We have modified the Abstract (lines 9-11 of the revised manuscript):

“Neuronal ensembles, coactive groups of neurons found in spontaneous and evoked cortical activity, are causally related to memories and perception, but it still unknown how stable or flexible they are over time.”

We also have modified the Introduction (lines 33-48 of the revised manuscript):

“Both stability and flexibility in different brain areas have been reported using single-cell statistics (Lütcke et al., 2013; Ziv et al., 2013; Driscoll et al., 2017; Gonzalez et al., 2019; Rule et al., 2019). […] Thus, we asked whether ensembles are preserved across days and how flexible they are, i.e., how many neurons firing together on one day continue to do so in the following days and how many of them stop firing together. Analyzing stable ensembles, we found that ~68 % of their neurons were preserved over weeks (stable neurons), whereas the rest were not (flexible neurons).”

2. There are claims in this paper about how ensembles 'implement long-term memories' in the introduction and conclusion and yet the authors never link the activity of ensembles to any behavioral or stimulus dependent feature. This language reaches far beyond the evidence provided in this paper. The introduction could provide some better framing for expectations of stability vs. drift in neural activity rather than focus on the link between ensembles and memory given that there isn't much focus on the ensembles' contribution to memory throughout. For example, the last sentence of the paper is not supported by data in the paper. Where is the link between ensembles and memory in the data? What is the evidence that transient ensembles are related to new or degraded memories? This reads as though it was the authors' hypothesis before doing the experiments and was not adjusted in light of the results.

We thank this opportunity to clarify this point. We now have clarified that this is just a hypothesis to motivate future studies of the possible relation between the long-term stability of ensembles and memories. We clarified the paragraphs in the abstract and discussion.

In the Abstract (lines 17-19 of the revised manuscript):

“Our results demonstrate that neuronal ensembles can last for weeks and could, in principle, serve as a substrate for long-lasting representation of perceptual states or memories.”

In the Discussion (lines 334-342 of the revised manuscript):

“The stronger internal functional connectivity of stable neurons could be mediated by short- or long-term synaptic plasticity (Carrillo-Reid et al., 2015b; Hoshiba et al., 2017) and may underlie the representation of memories. Since neuronal ensembles in the visual cortex have been associated with perceptual states or memories (Carrillo-Reid et al., 2019; Marshel et al., 2019), stable neuronal ensembles could represent long-term memories and transient ensembles could illustrate the emergence of new memories or the degradation of existing ones. Future experiments, perhaps using holographic optogenetics (Yang et al., 2018) during memory tasks, could test this hypothesis and explore the potential link between the stability of ensembles and the persistence of memories.”

We have also provided a better framing of the expectations in the Introduction (lines 29-40 of the revised manuscript):

“The optogenetic activation of the ensembles can lead to behavioral effects consistent with the hypothesis that they represent perceptual or memory states (Carrillo-Reid et al., 2019; Marshel et al., 2019). Interestingly, while single-cell tuning remains stable in visual cortex (Ranson, 2017; Jeon et al., 2018), a representational drift occurs across days (Deithch et al., 2020). Both stability and flexibility in different brain areas have been reported using single-cell statistics (Lütcke et al., 2013; Ziv et al., 2013; Driscoll et al., 2017; Gonzalez et al., 2019; Rule et al., 2019). However, there is a lack of multineuronal studies on cortical activity across days. Thus, we asked whether ensembles are preserved across days and how flexible they are, i.e., how many neurons firing together on one day continue to do so in the following days and how many of them stop firing together. We also explored whether the stability or flexibility of ensembles are different between spontaneous and visually evoked activity.”

3. There is no discussion around the alternative to stability of neuronal ensembles. What are the current theories about representational drift? For example, in Line 34 the authors present an expectation for stability without any reasoning for why there need not be stability. This lack of framing makes their job of explaining results in line 217 more difficult. There is a possibility that the most stable cells aren't more important – what is the evidence that they are? Does an ensemble need a core? Would be interesting to include some discussion on the possibility of a drifting readout (Line 223). [https://doi.org/10.1016/j.conb.2019.08.005]

We appreciate this suggestion. We have edited the discussion including current theories (lines 297-310 of the revised manuscript):

“Multiplane two-photon calcium imaging allowed tracking the same neurons and identifying neuronal ensembles across weeks. Stable ensembles were preserved across days, and 68% of their neurons continued to be active, while the rest of the neurons were replaced by new ones (Figure 4F). The consistent number of neurons preserving their interactions within ensembles across days does not support a representational drift in cortical responses at the single neuronal level (Discroll et al., 2017, Rule et al., 2019; Deithch et al., 2020). In fact, this could be precisely one of the functions of ensembles: to maintain a stable functional state in the midst of an ongoing homeostatic replacement of the activity of individual neuronal elements (Mrsic-Flogel et al., 2007; Lütcke et al., 2013; Hengen et al., 2013; Clopath et al., 2017). The stability of ensembles could be explained by the stability of dendritic spines (Yuste and Bonhoeffer, 2001). The weak connectivity of flexible neurons could be mediated by thin spines, which appear and disappear over days (Holtmaat et al., 2005), while the high connectivity of stable neurons could be maintained by thick spines, which last for months (Holtmaat et al., 2005; Grutzendler et al., 2002).”

4. How do activations in V1 in this dataset compare to other data collected from V1 while the animal is performing a task (where for example the angle of the gradings is relevant to how the mouse should respond)? I would be interested to know if the authors compared statistics of their ensembles to publicly available data recorded in V1 during a visually guided behavior. Are the ensembles tuned to anything in particular? Could they be related to movement? [http://repository.cshl.edu/id/eprint/38599/]

We appreciate these queries from the reviewer. We analyzed the publicly available data suggested by the reviewer (http://repository.cshl.edu/id/eprint/38599/), but we did not find any difference in the statistics with the behavior (evoked on day 1 “Churchland Lab dataset”, Figure 3 —figure supplement 5 and Figure 4 —figure supplement 2). Mice were recorded on different days and different regions, so we could not analyzed the same cells across days. So, further studies are needed to evaluate the stability of ensembles across days during the performance of a behavioral task.

We have added a paragraph in the Results (lines 243-257 of the revised manuscript):

“Furthermore, there were no significant differences when we analyzed the Churchland Lab dataset, where evoked activity was relevant for performing a task (Musall et al., 2019; Figure 3 — Figure Supplement 5 and Figure 4 — Figure Supplement 2). In all methods and datasets, the number of stable neurons within ensembles was above 60 % (Figure 4 — Figure Supplement 2B), the connection density of the stable neurons was above 0.5, which was significantly greater than the connection density of the lost neurons (Figure 4 — Figure Supplement 2D). In summary, we detected long-term ensemble stability regardless of the method or dataset used.”

We have added 2 extra supplement figures.

In addition, we also have performed ensemble analysis related with running speed and visual stimulation (Figure 3 – Supplement Figure 2-3). We observed that some ensembles are tuned to visual stimulus or locomotion, but the average change during locomotion or tuning of ensembles is near zero across days. Thus, the tuning and locomotion correlation of ensembles remain stable across days.

We have edited the next paragraph in the Results (lines 180-185 of the revised manuscript):

“In conclusion, stable ensembles cannot be explained by chance, and they were not only reactivated during the following days but also their correlation with locomotion and tuning remained unchanged across days (Figure 3 — Figure supplement 2-3). This speaks to the likelihood that neuronal ensembles are true functional circuit elements and not an epiphenomenon of the population activity or a statistical artifact.”

We have added 2 extra supplements.

5. The authors provide some hypotheses as to why fewer cells are active in the later imaging sessions (dead/dying cells?). This is worrisome in regards to how much it might have affected the imaged area's biology. One alternative hypothesis is that the animal is more familiar with the environment/ not running as much etc. Have the authors collected any behavioral data to compare over time?

We thank the reviewer for pointing this out. We recorded the running speed in most of our experiments. Contrary to the expectation, mice run more across days (Figure 1 — Figure Supplement 2B). At the same time, we have found a likely explanation for the decrease in active cells: we counted the number of neurons discarded due to poor signal to noise (< 18 dB), and we found a significant increase of discarded neurons only from day 1 to day 43-46 (24 ± 1% and 37 ± 4%, respectively; p = 0.034). Likely, the loss of active neurons is due to decrease in imaging quality (Figure 1 — Figure Supplement 2C) which in turn could be due to surgical-attachment-related microtraumas (Figure 1 — Figure supplement 3).

We added two supplementary figures.

We added the next paragraph in the Results (lines 93-101 of the revised manuscript):

“Differences in z-displacement were similar across days (< 4 µm, see Methods, Figure 1 — Figure Supplement 2A), and mice running speed tended to increase across days (Figure 1 — Figure Supplement 2B). Both factors could not utterly explain the decrease in the number of active neurons. Moreover, the percentage of discarded neurons, with poor signal to noise (PSNR < 18 dB), significantly increased from day 1 to day 43-46 (from 24 ± 1% to 37 ± 4%, mean ± SEM; p = 0.034; Figure 1 — Figure Supplement 2C). Probably, the loss of active cells on day 43-46 was due to a decrease in imaging quality, which could be partially explained by surgical-attachment-related traumas (Figure 1 — Figure supplement 3).”

6. How much do the results change when you vary the 50% threshold of preserved neurons within an ensemble (Line 146)? Does it make sense to call an ensemble stable when 50% of the cells change? Especially given that the cells analyzed as contributing to an ensemble are already sub-selected to be within the small population of stable cells (Line 119)?

We are grateful for these observations. We have performed the entire analysis by varying the threshold to define a stable ensemble from zero to one (0, 0.25, 0.33 —50 % of neurons—, 0.5, 0.75, and 1; Figure 3 — Figure Supplement 4; Figure 4 — Figure Supplement 1). There were fewer stable ensembles when we increased the threshold, as expected, but there were no significant increases in stable ensembles when we decreased this threshold, even to zero (Figure 3 — Figure supplement 4A). Regardless the threshold used, there are no differences in the functional structure of stable ensembles (Figure 3 — Figure supplement 4B-D), and they consistently preserve more than 60 % of their neurons (Figure 4 — Figure supplement 1B-D).

We have added in the Results (lines 190-199 of the revised manuscript):

“We asked if varying the threshold to define stability could change this result. If we set the strictest threshold (Jaccard similarity = 1, i.e., all the neurons remain in the same ensemble), we counted significantly less stable ensembles (P < 0.05; Figure 3 — Figure supplement 4A). On the other hand, we did not find any significant addition of stable ensembles when we reduced the threshold, even to zero (Figure 3 — Figure supplement 4A). This indicates that the maximum number of stable ensembles could be defined by the minimum number of ensembles found between two sessions. In all cases, regardless of the Jaccard similarity threshold used, there were no significant differences in the functional structure of the ensembles (Figure 3 — Figure supplement 4B-C). This result suggests that the functional structure of the ensembles is steady and independent of its stability.”

We have added two extra Supplementary Figures.

7. Cells are referred to as 'stable' when they're active on 3 different sessions that are separated in time. However, the authors find a smaller number of cells are stable over extended time (43-46 days later). If we extrapolate this over more time, would we expect these cells to continue to be stable? Given these concerns, it might make more sense to qualify the language around stability by the timespan over which these cells were studied.

We are not sure whether we fully understand what cells the reviewer is referring to, but we thank to the reviewer for this opportunity to discuss this point. The reviewer maybe referring to the common neurons (Figure 1K). In this case, we think that common neurons are reduced across days because we identified fewer active neurons (Figure 1H) and our recordings showed a displacement in the z-plane, likely losing some neurons and recording new ones (new figure showed in the response of concern 5 from this reviewer, Figure 1 — Figure Supplement 2A).

We have added a paragraph in the Discussion (lines 279-296 of the revised manuscript):

“We could not track the activity of every single neuron over weeks with sufficient signal to noise (PSNR > 18 dB, Figure 1H) since image quality decreased over days (Figure 1G). This could be due to the repeated experimental procedures on the same cortical location, decreased transgene expression, laser photobleaching, or surgical-attachment-related microtraumas (Figure 1 — Figure Supplement 3). Nevertheless, our method was sufficient to capture similar average activity from the neurons with adequate signal to noise over weeks (Figure 1I), agreeing with the hypothesis that cortical circuits maintain a basic homeostatic activity level, even in spite of perturbations (Mrsic-Flogel et al., 2007; Lütcke et al., 2013; Hengen et al., 2013; Clopath et al., 2017). However, we found that single-neuron responses were variable during the same repeated stimulation (Montijn et al., 2016; Stringer et al., 2019c). Indeed, the chance of finding any given neuron also active in a future session (”common neurons”) was less than 60 %, even 5 min later (Figure 1K; Tolias et al., 2007). This result is consistent with transient silencing of neurons (Prsa et al., 2017), which could be a neuronal correlate of the learning enhancement in deep neural networks (Srivastava et al., 2014; Rule et al., 2019). We found a continuous decrease in “common” neurons, which could be explained by the loss of neurons (Figure 1H) together with small z-plane displacement across days (Figure 1 — Figure supplement 2A). If one could maintain imaging quality and focus we would expect a consistent number of common neurons over weeks, regardless of the time recorded.”

8. Filtering frames to only coactive neurons for ensemble identification seems strange to me. Authors may be overestimating the extent of coactivation. What happens when you don't do this? How much do the results change when you don't subselect for Jaccard similarity? I would be interested to see how the results vary as you vary this threshold (Line 136).

We thank the reviewer for this suggestion. We have performed the entire analysis keeping all vectors (without selecting the most similar vectors). We added 2 supplementary figures (presented in the previous concern 4 by this reviewer; Figure 3 — Figure Supplement 5 and Figure 4 — Figure Supplement 2). In general, the statistics of ensembles are very similar. The only relevant difference was observed in the ensemble robustness, whose values were higher compared to the selection of most similar vectors and they were not significantly different between stable and transient ensembles. Without selecting vectors, the algorithm accommodates all vectors as ensemble activations, even if they are infrequent activations. This, by itself, increases the ensemble robustness, resulting in similarities between stable and transient ensembles during spontaneous activity. However, we could detected differences between stable and transient ensembles during evoked activity.

We added the following paragraph in the Results (lines 230-239 of the revised manuscript):

“In our study, we used all single sessions (“main”, 3 sessions/day/condition with a selection of vectors), and the results were similar when we used only one session per day (“single” 1 session/day/condition), all the raster without vector selection (“all vectors”), and the significant population coactivations (“coactivity peaks”; see Methods). The properties of the stable and transient ensembles as the neurons/ensemble and connection density remained with no differences (Figure 3 — Figure Supplement 5B-C), but ensemble robustness values had some variability between methods (Figure 3 — Figure Supplement 5D). However, the vector selection method proposed in this study differentiates significantly between stable and transient ensembles, specially during spontaneous activity.”

9. The term 'evoked activity' is misleading because the authors don't link these activations to the visual stimulus. There's no task, so the mice could be paying little attention to the stimulus. Should we really consider this activity to be visually driven? Could the authors provide any evidence of this?

We agree with the reviewer. There is no task in our experiments, and visual stimulation may be irrelevant to the mice. However, neuronal activity remains tuned to the visual stimulation across days. We have added one extra supplement during the stimulation sessions, which evidences the single-neuron tuning to the visual stimulation (Figure 1 —figure supplement 5).

We also have added a supplement figure in the concern 4 from this reviewer, which demonstrates the ensemble tuning to the visual stimulation (Figure 3 — Figure supplement 3).

10. A method like seqNMF could reveal ensembles that are offset in time. This looser temporal constraint could potentially reveal more structure. This should be run on the entire dataset (without stability sub-selection). I suggest this as a potential alternative or supplement to the method described by the authors. [https://elifesciences.org/articles/38471]

We agree with the reviewer. We have reanalyzed the data with the algorithm seqNMF as suggested by the reviewer, obtaining similar results. We have added a new supplementary figure showing the persistence of temporal sequences during spontaneous and evoked activity.

We now propose seqNMF as a complement of our method used in the Results (lines 249-252 of the revised manuscript):

“On the other hand, we also observed persistence across days of the temporal sequences of neuronal activations using the seqNMF toolbox (Mackevicius et al., 2019; Figure 4 — Figure Supplement 3), which we suggest in future studies to deeply analyze the dynamic of the temporal structure within ensembles.”

We also have added a new paragraph in the Methods (lines 561-567 of the revised manuscript):

“Finding neuronal sequences using the seqNMF toolbox. We used the seqNMF toolbox (https://github.com/FeeLab/seqNMF; Mackevicius et al., 2019) to detect possible neuronal patterns of sequential activation as an alternative to our method, which has a constraint to detect simultaneous neuronal activation (coactivation). We used the common neurons across days during spontaneous and evoked activity to find temporal sequences within windows of ~1 s (L = 12), a regularization parameter λ = 0.005, and 50 iterations (Figure 4 — Figure supplement 3).”

Again, we thank this reviewer for all detailed observations to strengthen our manuscript.

Reviewer #3 (Recommendations for the authors):Neuronal ensembles have been shown by this lab and others to constitute one basic functional unit for the representation of information in cortical circuits. It is therefore important to determine how stable these blocks of representation might be. If these ensembles were preserved across time and sensory stimuli, this would indicate a significant degree of structure underlying cortical representations. In a first attempt to address these important issues, this manuscript analyzes the long-term stability of ensembles of coactive neurons in the layer 2/3 of mouse visual cortex across several days. Ensembles were recorded during periods of spontaneous activity as well as during visual stimulation (evoked). For this, the authors record spontaneous and evoked activity using two-photon calcium imaging one, ten and 40 days after the first recording session. In order to maximize overlap between successive imaging sessions, the authors record three planes separated by 5 microns almost simultaneously (9ms interval) using an electrically-tunable lens. They show that ensembles extracted during visual stimulation periods are more stable on days 2 and 10 than those computed during spontaneous activity. Stable ensembles display a higher "robustness" (a parameter that quantifies how many times a given ensemble is repeated and how similar these repeats are). Neurons displaying stable membership are more functionally connected than unstable ones. It is concluded that such observed stability of spontaneous and evoked ensembles across weeks could provide a mechanism for memories. Long-term calcium imaging within the same population of neurons is a real challenge that the authors seem to overcome in the study. The conclusions are important, my main concern relates to the number of experiments and analyses supporting these findings as detailed below.

We are very thankful to this reviewer for the useful comments on our manuscript.

Number of experiments and statistics: According to Table 1, two mice with GCamP6f have been through the complete imaging protocol (days 1,2, 10 and 43) but none with the 6s, since 3 missed the intermediate measure (day 10) and one the last point (day 40+). Therefore five mice have been recorded over weeks with two different indicators, but only two were sampled on day 10. One mouse was only recorded until day 10. Altogether, this is quite a low sampling, but the experiments are certainly difficult. However, the total number of experiments analyzed is higher, due to the repeat of 3 sessions on the same mouse per day. This certainly contributes to reaching significance. However, the three samples from the same mouse are not independent points. Are the FOVs different for each session in the same mouse? If they are the same, then the statistics should be repeated but treating all experiments from the same mouse as single experiments. I would suggest repeating the analysis but using only one data point per mouse per day. Also, given that two different indicators were used (6s and 6f), one would need to see whether the statistics are the same in the two conditions.

We appreciate the acknowledgement of the difficulty of the experiments. The fields of view are the same for each animal, so we now compute the statistics (mean ± SEM) using a single data point per mouse per day (“single (1 session/day)”, Figure 3 — Figure supplement 5 and Figure 4 — Figure supplement 2). To increase the numbers, we also have analyzed two publicly datasets available (Figure 3 — Figure supplement 5 and Figure 4 — Figure supplement 2): seven GCaMP6f mice recorded in 3 different days from The Allen Brain Institute (“Allen Brain Institute dataset”) and four GCaMP6f mice recorded in the same day during a visually guided behavior (“Churchland Lab dataset”). We also have separated the results for each indicator used (“only GCaMP6f” and “only GCaMP6s” in Figure 3 — Figure supplement 5 and Figure 4 — Figure supplement 2). In all datasets, results were similar.

We have added in the Results (lines 230-257 of the revised manuscript):

“In our study, we used all single sessions (“main”, 3 sessions/day/condition with a selection of vectors), and the results were similar when we used only one session per day (“single” 1 session/day/condition) It should be noted that the results were heterogeneous among GCaMP6s and GCaMP6f mice, consistent with previous studies (Musall et al., 2019). We analyzed four GCaMP6s and two GCaMP6f mice in our dataset, seven GCaMP6f mice from the Allen Brain Institute dataset, and four GCaMP6f mice from Churchland Lab dataset. In summary, we detected long-term ensemble stability regardless of the method or dataset used.”

We have added two supplementary figures.

Robustness: the authors compute this metric, as the product of ensemble duration and average of the Jaccard similarity and find that stable ensembles display higher robustness: isn't it expected that robustness is higher in stable ensembles given that stable ensembles should be observed more often?

We apologize for our lack of clarity. As it is an independent measure, it is not necessarily expected that robustness should be higher in stable ensemble. We can compute the robustness value in one single session, but, independently, we can identify if an ensemble is stable when it is observed in a future session. For example, two ensembles could have the same robustness value in the first session, but only one of them is active in a future session.

We have clarified the following paragraph in the Methods (lines 517-522 of the revised manuscript):

“Ensemble robustness, we introduced here as robustness = similarity · activity, where similarity is the average of the Jaccard similarity between every pair of column vectors of the ensemble matrix raster Ej, and activity is the fraction of ensemble active frames to the total frames of the session. The higher the value, the higher the robustness. We computed ensemble robustness for every single session, so it would not be expected beforehand if the ensemble would be stable or not.”

We add the following text in the legend of Figure 3A:

“Note that ensemble robustness is computed per ensemble per single session.”

Evoked ensembles: It seems to me that evoked ensembles are ensembles extracted during continuous imaging periods that include stimulation. However, one would expect evoked ensembles to be the cells activated time-locked to the visual stimulation. This notion only appears at the end of the paper with "tuned" neurons in Figure 4. In the discussion, authors conclude lines 205-207 that "sensory stimulus reactivate existing ensembles". I do not think this is supported by the analysis performed here. For this, I believe that one would need to compare, within the same mouse the amount of overlap between spontaneous ensembles and "tuned neurons".

Thank you for pointing this out. We have added the analysis of ensembles during spontaneous activity which also were active during visually evoked activity and vice versa (see “S day 1 vs E day n | E day 1 vs S day n” on Figure 3 — Figure Supplement 5 and Figure 4 — Figure Supplement 2). We also have counted the fraction of tuned neurons found across days that overlapped with the maximally matched spontaneous ensemble found on day 1. In addition and consistent with this result, we cite previous work by Miller et al., (2014).

We have added the following paragraph in the Discussion (lines 271-278 of the revised manuscript):

“Somewhat surprisingly, we did not find any relevant differences in long-term stability between spontaneous and visually evoked activity. Indeed, we found that ensembles during spontaneous activity were also active during visually evoked activity and vice versa (see “S day 1 vs E day n | E day 1 vs S day n” on Figure 3 — Figure Supplement 5 and Figure 4 — Figure Supplement 2). In fact, more than 60 % of neurons from spontaneous ensembles on day 1 were found to be tuned to a specific stimulus on following weeks. This result is consistent with the hypothesis that sensory stimuli reactivate existing ensembles, which are already present in the spontaneous activity (Miller et al., 2014).”

How representative are the illustrated examples in Figures 2 and 3? The authors report that about 20 neurons remain active from day 1 to 46 but their main figures display example rasterplots with more than 60 neurons, which is three times more than the average. Is this example representative? Which indicator was used? Is there a difference in stability between 6f and 6s?

We thank this opportunity to clarify these figures. We previously showed an example between common neurons from day 1 and 2 in the Figures 2-3, but now we have modified them showing a representative example of the common neurons between days 1 and 43 from a GCaMP6s mouse. So, we now display a representative example from one mouse GCaMP6s in Figures 2-3 and one GCaMP6f in Figure 4. It seems there are no differences in stability between GCaMP6s and GCaMP6f as we showed above in response to a previous concern by this reviewer (Figure 3 — Figure supplement 5 and Figure 4 — Figure supplement 2).

We have edited Figures 2 and 3.

Rasterplot filtering: The authors chose to restrict their ensemble analysis to frames with "significant coactivation". Why not use a statistical threshold to determine the number of cells above which a coactivation is significant instead of arbitrarily setting this number to three coactive neurons? In cases of high activity this number may be below significance.

We appreciate this observation. We have added a new analysis of the significant population coactivation peaks, and the results of stability remain similar. The results are now shown in two supplementary figures, added in a previous response to this reviewer (“coactivity peaks” in Figure 3 — Figure supplement 5 and Figure 4 — Figure supplement 2). We added the following paragraph in the Methods (lines 477-484 of the revised manuscript):

“Detection of coactivity peaks to identify ensembles. Alternatively, we analyzed our dataset using a method to extract neuronal ensembles based on significant population coactivity (Pérez-Ortega et al., 2016). Briefly, we obtained a 1 × F vector, where F is the number of frames, by summing the coactive neurons from the raster matrix E. Then, we perform 1,000 surrogated raster matrices, by randomly circular shifting in time the activity of every single neuron and computing the coactivity given by chance. We determined a significant coactivity threshold (P < 0.05) from surrogated coactivity, and the vectors above this threshold were clustered to extract neuronal ensembles.”

We have added a paragraph to the Results (lines 230-239 of the revised manuscript):

“In our study, we used all single sessions (“main”, 3 sessions/day/condition with a selection of vectors), and the results were similar when we used only one session per day (“single” 1 session/day/condition), all the raster without vector selection (“all vectors”), and the significant population coactivations (“coactivity peaks”; see Methods). The properties of the stable and transient ensembles as the neurons/ensemble and connection density remained with no differences (Figure 3 — Figure Supplement 5B-C), but ensemble robustness values had some variability between methods (Figure 3 — Figure Supplement 5D). However, the vector selection method proposed in this study differentiates significantly between stable and transient ensembles, especially during spontaneous activity.”

Demixing neuronal identity: The authors assign a neuron to an ensemble if it displays at least a functional connection with another neuron. They use reshuffling to test significance of functional links but still it seems that highly active neurons are more likely to display a high functional connectivity degree and therefore to be stable members of a given ensemble with that definition of ensemble membership. What is the justification to define membership based on pairwise functional connectivity? The finding that core ensemble members display a high functional degree may be just a property reflecting a property of highly active neurons (as previously described by Mizuseki et al., 2013).

We appreciate this query. We compute the percentage of activity versus the percentage of functional connections of 297 and 314 common neurons during spontaneous and evoked activity, respectively, from all the mice in our dataset. The coefficient of determination was near zero (R2 = 0.001), so there is no relation between neuron’s activity and its significant functional connections. We have added a new supplement (Figure 2 — Figure supplement 1).

We have added this line in the Results (lines 138-139 of the revised manuscript):

“The functional connections of each neuron were independent of its level of activity (R = 0.001, Figure 2 — Figure Supplement 1)”

Type of neurons imaged: The authors use Vglut1-Cre mice, therefore they are excluding GABAergic cells from their study, this should be clearly mentioned and even discussed.

Thanks for pointing this out. We now clarify that we recorded the activity of pyramidal neurons, excluding GABAergic neurons. Additionally, we acknowledge that this is a limitation of this study. We have modified the following paragraph in the Results (lines 56-59 of the revised manuscript):

“We performed two-photon calcium imaging of pyramidal cells in layer 2/3 of the visual cortex from six transgenic mice (GCaMP6s, n = 4 animals; and GCaMP6f, n = 2) through a cranial window to examine the stability of ensembles under visually-evoked and spontaneous activity.”

We have added in the Discussion the following paragraph (lines 311-320 of the revised manuscript):

“One limitation of this study is that we did not image the activity of GABAergic interneurons, which are at least 28 different types based on the morphoelectric and transcriptomic classifications (Yuste et al., 2020; Gouwens et al., 2020; Yao et al., 2021). Parvalbumin (PV) interneurons would stabilize the cortical circuit while somatostatin (SOM) and vasoactive intestinal peptide (VIP) interneurons would modulate the gain of pyramidal neurons according to recent models (Bos et al., 2020; Millman et al., 2020). Moreover, single-neuron statistics showed that PVs in the visual cortex undergo faster homeostasis (Hengen et al., 2013) and are more stable than pyramidal cells (Ranson, 2017). Further studies are needed to evaluate the stability within interneuron interactions and the interactions between interneurons and pyramidal neurons.”

Volumetric imaging: I am not sure one can say that "volumetric imaging" was performed here, rather this is multi-plane imaging.

We have made the change from “volumetric imaging” to “multiplane imaging” thorough out the manuscript.

Mouse behavior: there is little detail concerning mouse behavior, are mice allowed to run? What is the correlation between ensemble activation and running?

Thank you for this observation. We have performed the correlation between ensembles and the running speed of the mice during spontaneous and visually evoked activity. We have added two new supplementary figures (Figure 3 — Figure supplement 2 and Figure 3 — Figure supplement 3).

Abstract: the authors should say that 46 days is the longest period they have been recording, otherwise it gives the wrong impression that after 46 days ensembles are no longer stable. Also "most visually evoked ensembles" should be replaced by "ensembles observed during periods of visual stimulation" (see above). "In stable ensembles most neurons still belonged to the same ensemble after weeks": how could ensembles be stable otherwise?

We appreciate the suggestions. We have modified the Abstract to:

“Neuronal ensembles, coactive groups of neurons found in spontaneous and evoked cortical activity, are causally related to memories and perception, but it still unknown how stable or flexible they are over time. We used two-photon multiplane calcium imaging to track over weeks the activity of the same pyramidal neurons in layer 2/3 of the visual cortex from awake mice and recorded their spontaneous and visually-evoked responses. Less than half of the neurons were commonly active across any two imaging sessions. These “common neurons” formed stable ensembles lasting weeks, but some ensembles were also transient and appeared only in one single session. Stable ensembles preserved ~68 % of their neurons up to 46 days, our longest imaged period, and these “core” cells had stronger functional connectivity. Our results demonstrate that neuronal ensembles can last for weeks and could, in principle, serve as a substrate for long-lasting representation of perceptual states or memories.”

We have added this paragraph in the Discussion (lines 323-329 of the revised manuscript):

“While there is a significant state of flux in cortical activity at any given moment, there is a subset of neurons that remain active through weeks and form neuronal ensembles. The stability of ensembles that we report could be an underestimated since we only measured snapshots of cortical activity. These stable ensembles, which also have some rotation of their individual neuronal components, appear anchored by a core of neurons. Specifically, 68 % of neurons remain consistently active within ensembles up to 46 days later (Figure 4F) and had stronger functional connectivity (Figure 4H).”

Discussion: I found the discussion quite succinct. It lacks discussing circuit mechanisms for assembly stability and plasticity (role of interneurons for example?), limitations and possible biases in the analysis and placing results in the perspective of other studies analyzing the long-term stability of neuronal dynamics.

Thank you for this suggestion. We have extended the discussion, comparing our results with recent longitudinal studies and also discussing representational drift theories about the stability and flexibility of the cortical activity.

We have added the following paragraphs in the Discussion (lines 283-310 of the revised manuscript):

“Nevertheless, our method was sufficient to capture similar average activity from the neurons with adequate signal to noise over weeks (Figure 1I), agreeing with the hypothesis that cortical circuits maintain a basic homeostatic activity level, even in spite of perturbations (Mrsic-Flogel et al., 2007; Lütcke et al., 2013; Hengen et al., 2013; Clopath et al., 2017). However, we found that single-neuron responses were variable during the same repeated stimulation (Montijn et al., 2016; Stringer et al., 2019c). Indeed, the chance of finding any given neuron also active in a future session (”common neurons”) was less than 60 %, even 5 min later (Figure 1K; Tolias et al., 2007). This result is consistent with transient silencing of neurons (Prsa et al., 2017), which could be a neuronal correlate of the learning enhancement in deep neural networks (Srivastava et al., 2014; Rule et al., 2019).

Multiplane two-photon calcium imaging allowed tracking the same neurons and identifying neuronal ensembles across weeks. Stable ensembles were preserved across days, and 68% of their neurons continued to be active, while the rest of the neurons were replaced by new ones (Figure 4F). The consistent number of neurons preserving their interactions within ensembles across days does not support a representational drift in cortical responses at the single neuronal level (Discroll et al., 2017, Rule et al., 2019; Deithch et al., 2020). In fact, this could be precisely one of the functions of ensembles: to maintain a stable functional state in the midst of an ongoing homeostatic replacement of the activity of individual neuronal elements (Mrsic-Flogel et al., 2007; Lütcke et al., 2013; Hengen et al., 2013; Clopath et al., 2017). The stability of ensembles could be explained by the stability of dendritic spines (Yuste and Bonhoeffer, 2001). The weak connectivity of flexible neurons could be mediated by thin spines, which appear and disappear over days (Holtmaat et al., 2005), while the high connectivity of stable neurons could be maintained by thick spines, which last for months (Holtmaat et al., 2005; Grutzendler et al., 2002).”

We thank again this reviewer for the constructive comments to our manuscript.